# The Piezo channel is a mechano-sensitive complex component in the mammalian inner ear hair cell

Jeong Han Lee[1,10], Maria C. Perez-Flores [1,10], Seojin Park [1,2], Hyo Jeong Kim[1], Yingying Chen [1], Mincheol Kang [1,2], Jennifer Kersigo[3], Jinsil Choi[1], Phung N. Thai [4], Ryan L. Woltz [4], Dolores Columba Perez-Flores[1], Guy Perkins[5], Choong-Ryoul Sihn[1], Pauline Trinh[4], Xiao-Dong Zhang[4], Padmini Sirish [4], Yao Dong[6], Wayne Wei Feng [6], Isaac N. Pessah[6], Rose E. Dixon [7], Bernd Sokolowski [8], Bernd Fritzsch [3], Nipavan Chiamvimonvat[4,9] & Ebenezer N. Yamoah [1] ✉

The inner ear is the hub where hair cells (HCs) transduce sound, gravity, and head acceleration stimuli to the brain. Hearing and balance rely on mechanosensation, the fastest sensory signals transmitted to the brain. The mechanoelectrical transducer (MET) channel is the entryway for the sound-balance-brain interface, but the channel-complex composition is not entirely known. Here, we report that the mouse utilizes Piezo1 (Pz1) and Piezo2 (Pz2) isoforms as MET-complex components. The Pz channels, expressed in HC stereocilia, and cell lines are co-localized and co-assembled with MET complex partners. Mice expressing non-functional *Pz1* and *Pz2* at the ROSA26 locus have impaired auditory and vestibular traits that can only be explained if the Pzs are integral to the MET complex. We suggest that Pz subunits constitute part of the MET complex and that interactions with other MET complex components yield functional MET units to generate HC MET currents.

Transducer ion channels are the sensory modalities' gateway to the nervous system. The activation of transducer channels generates receptor potentials in sensory receptors, which are converted directly into primary sensory neural codes. The identity of the mechanoelectrical transducer (MET) channel complex responsible for detecting sound, gravity, and head acceleration in inner ear hair cells (HCs) remains an enigma[1,2]. The MET complex is housed in an HC actin-based stereocilium[3] located at the tips[4] of shorter stereocilia connected to the side of longer stereocilia by a tip link[5–7] that gates the

MET channels[8–10]. In the tip-link's absence, an anomalous HC MET current can be invoked from the cuticular plate membrane[11,12]. Mechanical deflection towards the tallest stereocilia exerts tension in the tip link, transmitting force directly onto the MET channel to increase the open probability ($P_o$), while deflection towards the shortest stereocilia causes slack in the tip link to reduce channel $P_o$[13,14]. Biophysical and pharmacological features of the MET channels include sub-millisecond activation time constants[15,16], weak cation-selectivity, and large molecule permeability, such as the fluorescent dye FM1-

[1]Department of Physiology and Cell Biology, School of Medicine, University of Nevada, Reno, NV 89557, USA. [2]Prestige Biopharma, 11-12F, 44, Myongjigukje7-ro, Gangseo-gu, Busan 67264, South Korea. [3]Department of Biology, University of Iowa, Iowa City, IA, USA. [4]Division of Cardiovascular Medicine, Department of Internal Medicine, University of California, Davis, CA 95616, USA. [5]National Center for Microscopy and Imaging Research, University of California San Diego, La Jolla, CA 92093, USA. [6]Department of Molecular Biosciences, School of Veterinary Medicine, University of California, Davis, 1089 VM3B, Davis, CA 95616, USA. [7]Department of Physiology & Membrane Biology, Tupper Hall, One Shields Avenue, Davis, CA 95616, USA. [8]Department of Otolaryngology-Head and Neck Surgery, Morsani College of Medicine, University of South Florida, Tampa, FL 33612, USA. [9]VA Northern California Healthcare System, Sacramento, USA. [10]These authors contributed equally: Jeong Han Lee, Maria C. Perez-Flores. ✉e-mail: enyamoah@gmail.com

43[17–19], and aminoglycoside antibiotic[20]. Accordingly, the MET channel has a sizeable unitary conductance of ~100-pS[21].

Considerable evidence has been used to identify components of the tip link and candidate protein subunits constituting the MET complex, but the molecular composition remains unclear. Experiments have shown that the tip link comprises cadherin 23 (Cdh23) at the upper end and protocadherin15 (Pcdh15) at the lower end of the filamentous structure, whereby the two proteins form cis-homodimers that interact in trans via their opposing N-termini[22–24]. The transmembrane inner ear (Tmie) protein[25] and the Ca²⁺/integrin binding family member 2 (Cib2) protein[26,27] may be coupled to the MET channel, specifically to the tip link and the cytoskeleton. The Lipoma HMGIC fusion partner-like 5 protein (Lhfpl5)[28] likely serves as an allosteric modulator of the MET channel. The prevalent promising MET channel candidates have been the transmembrane channels 1 and 2 (Tmc1 and Tmc2)[29,30]. Akin to other components of the MET complexes, Tmc1 and Tmc2 are localized at stereocilia tips[31], and their onset in HC expression coincides with the beginning of the MET current[32]. Multiple mutations of the *Tmc1* allele yield hearing loss, which alters MET channel conductance and Ca²⁺ permeability[33]. However, the association between genetic mutations and hearing loss is not unique to the *Tmc* alleles and are also found among candidates for the MET complex[34,35]. Tmc has structural similarities to the Ca²⁺-activated Cl⁻ channel (TMEM16A)[29], and Tmc orthologs may constitute a channel[36] that is mechanically sensitive in liposomes[37]. However, other attempts have failed to demonstrate the Tmcs as ion channels[38–40].

Piezo1 (Pz1) and Piezo2 (Pz2) are mechanically-sensitive ion channels[41,42] that respond to shear stress and pressure changes in blood vessels and the bladder[43] and are touch-sensitive in Merkel cells[44]. Pz channels are trimeric assemblies in which each modular subunit consists of a bowl-shaped monomer with a central pore, an extracellular cap, and three curved intracellular beams, generating a lipid curvature[45–47]. Previous reports did not detect Pz1 in the stereocilia of HC[48], but Pz2 functions may account for the anomalous reversed polarity current (RPI)[12]—null deletion of *Pz2* aborts high-frequency hearing in mice[49].

Here, we show that HCs express *Pz1* and *Pz2* transcripts and proteins, and the two proteins co-localize at the tips and sides of stereocilia and cuticular plate membranes with and without Tmc1 and Tmc2 co-assembly. Therefore, despite reports regarding Pz1, our results demonstrate colocalization of Pz1 and Pz2 at the HC stereocilia transduction hub in the cochlea and vestibular end-organs. Although sequence homology between Pz1 and Pz2 is ~42%[50,51], analyses of the trimeric interface suggest ~84% homology, providing the possibility for hetero-trimeric interaction. We generated non-functional *Pz1* and *Pz2* knockin (*ki*) mouse models, floxed at the ROSA26 locus. Results demonstrate mutant (*ki, mu*) *Pz1* and *Pz2*, but not wildtype *Pz1* under two different HC-specific Cre-recombinase, Myosin 15 (*Myo15 (mc)*) and calretinin (*Calb2 (cc)*), cause hearing and vestibular dysfunction. Co-immunoprecipitation analyses using cochlear tissue and cell lines expressing Tmc- and Pz-fusion proteins revealed the co-assembly of Pz and Tmc and coupling, using fluorescence resonance energy transfer (FRET) in situ and in vitro. Pz is a part of the MET complex and interacts with multiple proteins, including Pcdh15, Tmc1, Lhfpl5, Tmie, and Cib2, suggesting a role in the MET complex assembly. We propose the MET complex includes multimers of the Pz channel subunits that serve along with other regulatory and indispensable binding partners such as Tmc to confer a functional MET unit.

## Results

### Pz1/2 and Tmc1/2 are expressed in cochlear inner and outer hair cells (IHCs and OHCs) and vestibular organ HCs

To quantify the relative abundance of candidate MET channel transcripts in HCs, we used single-molecule fluorescent hybridization (smFISH) to determine the expression of *Tmc1*, *Pz1*, and *Pz2* transcripts in postnatal (P) day 10–12 (P10-P12) cochlear sections. Because the present experiments were restricted to hearing onset, and *Tmc2* transcript levels are undetectable after P8[52], *Tmc2* mRNA was not evaluated. The total number of RNA molecules detected per inner and outer hair cell (IHC/OHC) was calculated across independent replicates (Fig. 1a–c). *Tmc1* (in green, encoding Tmc1) was richly expressed in IHC and OHC, followed by *Pz2* and *Pz1* (in red, encoding Pz2 and Pz1). Negative and positive controls are shown (Supplement Fig. 1 (S1)). Compared with *Pz1 and Pz2*, *Tmc1* transcript levels are ~7-fold greater than *Pz* in HCs (Fig. 1c).

To validate the Pz1/2 expression in HCs, we used *Pz1-tdT: Pz2-GFP* mice, expressing Pz1 and Pz2 fusion proteins with tdTomato (tdT) and green fluorescence proteins (GFP), respectively[44]. We identified tdT-Pz1 (in red) and GFP-Pz2 (in green), co-localized in IHC and OHC stereocilia tips and sides of the whole-mount cochlear epithelium (Fig. 1d–f, counterstained with phalloidin for actin (in blue). Co-localized Pz1-tdT and Pz2-GFP puncta were observed in HC stereocilia in whole-mount utricle (Fig. 1g). The inter-channel (Pz1-Pz2 and Pz2-Pz1) nearest neighbor distances (NND) distributions are summarized in Fig. 1h, i. There was no statistical difference between NND (Pz1-Pz2) and NND (Pz2-Pz1) in IHC and OHCs. The mean NND (Pz1-Pz2) was $0.12 \pm 0.04$ μm ($n = 16$, data from 5 mice). A bimodal distribution, with first and second modes at $0.07 \pm 0.03$ μm and $0.35 \pm 0.06$ μm ($n = 16$), was calculated and obtained from a finite mixture of two Gamma functions, suggesting a relationship (either functional or spatial) between the two channels.

The Pz channel consists of trimeric subunits[45]. Pz1 and Pz2 sequences showed ~42% homology[50,51]. In contrast to the full-length sequence, the identified Pz1 subunit interface revealed 75% identical and 84% conserved residues with Pz2 (S2–3), which is similar to the heteromeric SK-channel subunit interacting interface that occurs in vitro and in vivo[53], raising the possibility for hetero-trimeric inter-action between Pz1 and Pz2 (see Method for detailed analyses). Results from proximity ligation assays (PLA) also affirmed at least ~40-nm closeness between Pz1 and Pz2 in HCs (S4). The spatial propinquity of the two-channel subunits in HC stereocilia and cuticular plate suggested a functional interaction.

We reasoned that if HC-Pz subunits constitute components of the MET complex, they should localize with Tmc subunits, components of the MET complex[30]. We confirmed this prediction using *Tmc2-AcGFP: Pz1-tdT* and *Tmc1-mCherry: Pz2-GFP* mice[31]. At P10-P12,Tmc2-AcGFP and Pz1-tdT expression was detected in IHCs, OHCs, and vestibular HC stereocilia (Fig. 2a–c, e). Co-localized and single-expression of Tmc2 and Pz1 was also detected at the level of the cuticular plate membrane. Figure 2d shows Pz2-GFP (green) and Tmc1-mCherry (red) in OHC and IHC stereocilia tips. Serial z-sections revealed colocalization of Tmc1 and Pz2 at stereocilia tips. The Pz1-Tmc2 and Pz2-Tmc1 NND distribution was ~$0.13 \pm 0.07$ μm ($n = 16$, data from 5 mice) in IHC stereocilia and ~$0.14 \pm 0.09$ μm ($n = 16$, data from 5 mice) in OHC stereocilia (Fig. 2f–h). At the cuticular plate, segregation was apparent between the two proteins (S5). While these results showed Pz was located at the presumed MET-current generation site, we needed to test whether the response properties of the Pz-current resemble the regular or the anomalous HC MET current, assuming both currents are derived from similar channel complexes.

### Properties of the Pz-channel current resemble the HC anomalous MET current

We overexpressed mouse *Pz1* (*mPz1*) plasmid in Neuro2A (N2A) cells. Membrane localization revealed by total internal reflection fluorescence (TIRF) microscopy accounted for ~13% of mPz1-mClover3 fusion protein expression. In contrast, ~8% of mTmc1-mRuby3 fusion protein membrane expression was measured 24 h post-transfection (S6). After 24–48 h of mPz1 plasmid transfection and mechanically displacing the

N2A cell body membrane via a piezo-electric-driven probe, whole-cell recordings at −80 mV revealed a Pz1 current ($I_{Pz1}$). The maximum $I_{Pz1}$ amplitude ranged from 0.7 to 6 nA, while the mean $I_{Pz1}$ was $3 \pm 1$ nA ($n = 17$; (S7a)). In contrast, endogenous mechanically-activated (MA) current in N2A cells ranged from 0.03–0.2 nA, with a mean value of $0.10 \pm 0.04$ nA ($n = 44$), in keeping with an earlier report[41]. The current ($I_{Pz1}$)-displacement relationship was fitted with a two-state Boltzmann function with half-activation ($X_{1/2}$) $= 0.8 \pm 0.1 \mu m$ and displacement sensitivity $X_{1/2}$ at $= 61.1 \pm 6.2$ pA $\mu m^{-1}$ ($n = 17$; S7b). The $I_{Pz1}$ was activated with a constant displacement and stepped to varying voltages. The resulting $I_{Pz1}$ traces are shown as an inset (S7c). The instantaneous current-voltage relation yielded a linear function between −120 and +90 mV. The reversal potential was $−7 \pm 5$ mV ($n = 14$) (S7c), consistent with a nonselective cation channel and compatible with the HC MET current[42,54].

Hair cell MET current displays a rectification mechanism, suggesting a reduced pore size at positive voltages[55]. We used small-amine molecules as permeant ions[56] to examine the channel-pore shape by plotting the current-voltage relation of the relative $I_{Pz1}$ at −90 and −120 mV (S7d, e). The channel inner and outer radii were estimated using reported empirical fitting analyses by plotting the amine radius vs. the relative current at −120 mV[56,57]. An inner-face pore radius of $7 \pm 4$ Å and an outer-face radius of $20 \pm 5$ Å was estimated (S7f, g). The findings align with HC MET channel[53] estimations and the upward concave structure of Pz[47,56]. Next, the $Ca^{2+}$-dependent $I_{Pz1}$ decay was evaluated with varying concentrations of patch-pipette $Ca^{2+}$ chelators, EGTA, and BAPTA (S8a–e). At −80 mV, where the driving force for $Ca^{2+}$ is greater, with 5-mM pipette EGTA, $I_{Pz1}$ decay was rapid (decay time constant ($\tau_{decay}$) at maximum $I_{Pz1} = 24 \pm 3$ ms ($n = 15$)). In contrast, at 70 mV, the $\tau_{decay}$ of $I_{Pz1}$ recorded with patch-pipette-BAPTA, a more

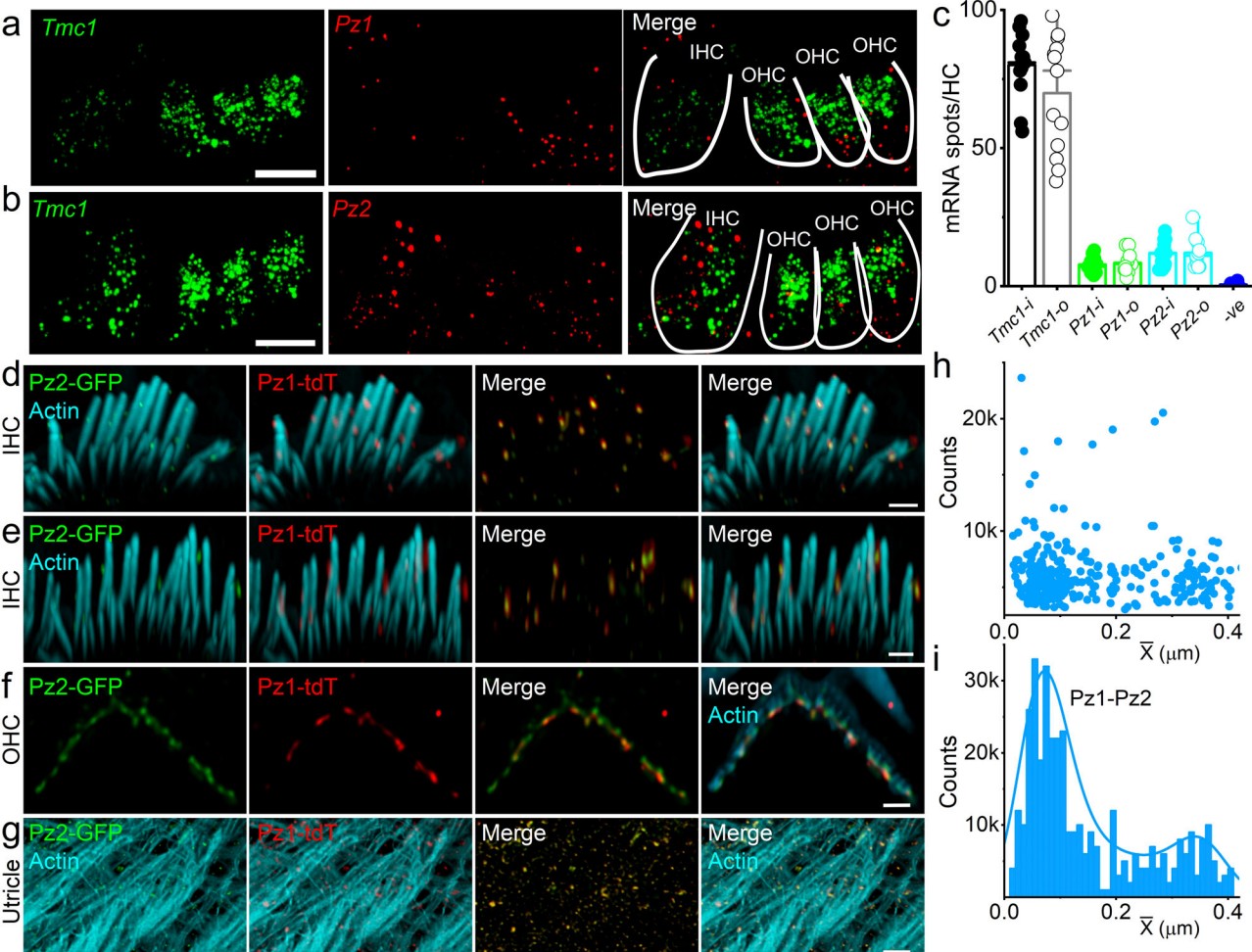

**Fig. 1 | smFISH localizes transcripts encoding *Tmc1*, *Pz1*, and *Pz2* in the IHCs and OHCs.** The expression of *Pz* and *Tmc* transcripts in HCs, using smFISH on the organ of Corti (OC) sections, P12 mice. **a, b** Tmc1 RNA molecules (*Tmc1*) (green), Pz1 (*Pz1),* and Pz2 (*Pz2*) (red) were detected as fluorescent puncta in HCs. Outlines of a single row of IHCs and three OHCs, marked in white. Negative and positive controls are shown in Supplement Fig. 1 (S1). **c** The mean number of RNA molecules per HC was calculated. *Tmc1* was most abundantly expressed (mean, IHC, noted as i, $80 \pm 3$ ($n = 13$), OHC, as o, $70 \pm 6$ ($n = 13$). *Pz1* and *Pz2* expression in IHCs (i) and OHCs (o) was ($Pz1i$, $8 \pm 1$ ($n = 11$), $Pz1o$, $9 \pm 1$ ($n = 11$), $Pz2i$, $12 \pm 2$ ($n = 11$), $Pz2o$, $12 \pm 2$ ($n = 11$), and negative probe, $0.4 \pm 0.3$ ($n = 13$) (S1). Data show individual replicates (animals) and mean ± SEM. There were significant differences at the $p < 0.05$ level for tested probes (*Tmc1*, *Pz1*, and *Pz2*) and negative controls F(6,74) = (128) $p = 5.0 \times 10^{-37}$. *Post hoc* comparisons using the Tukey HSD test indicate that $Pz1o$ vs $Tmc1i$ ($p = 1 \times 10^{-22}$); $Pz1i$ vs $Tmc1o$ ($p = 1 \times 10^{-25}$); $Pz1o$ vs $Tmc1i$ ($p = 1 \times 10^{-23}$); $Pz1o$ vs $Tmc1o$

($p = 2.1 \times 10^{-24}$); $Pz2i$ vs $Tmc1i$ ($p = 3.4 \times 10^{-26}$); $Pz2i$ vs $Tmc1o$ ($p = 4.4 \times 10^{-21}$); $Pz2o$ vs $Tmc1i$ ($p = 6.1 \times 10^{-23}$); $Pz2o$ vs $Tmc1o$ ($p = 1.8 \times 10^{-24}$) are significantly different. All experiments were repeated in n animal samples, and at least 15 images were collected, sampled blindly by three individuals, and averaged for each independent experiment. Scale bar = 10 μm (**a**, **b**). **d**, **e** GFP, and tdT fluorescence, visualized in the whole-mount cochlea from a cross between $Pz1^{tdT}$ and $Pz2^{GFP}$ ($Pz1^{tdT}/Pz2^{GFP}$) in P10 mice. Stereocilia were counterstained with phalloidin (in blue). Expression of Pz1 and Pz2 are co-localized in IHC (**d**, **e**), OHC (**f**), and utricular stereocilia (**g**). Scale bar (**d–f** 1 μm, **g** 2 μm). **h**, **i** a scatter plot and a histogram depict the counts and frequency of NND (Pz1-Pz2) in HC stereocilia. The bimodal distribution of NND suggests a non-random functional or spatial relationship between the two channel isoforms. At least 5 animals in each group were tested. Images were collected from 16 cochlear sections. Source data (**c**, **h**, **i**) are provided as a Source Data file.

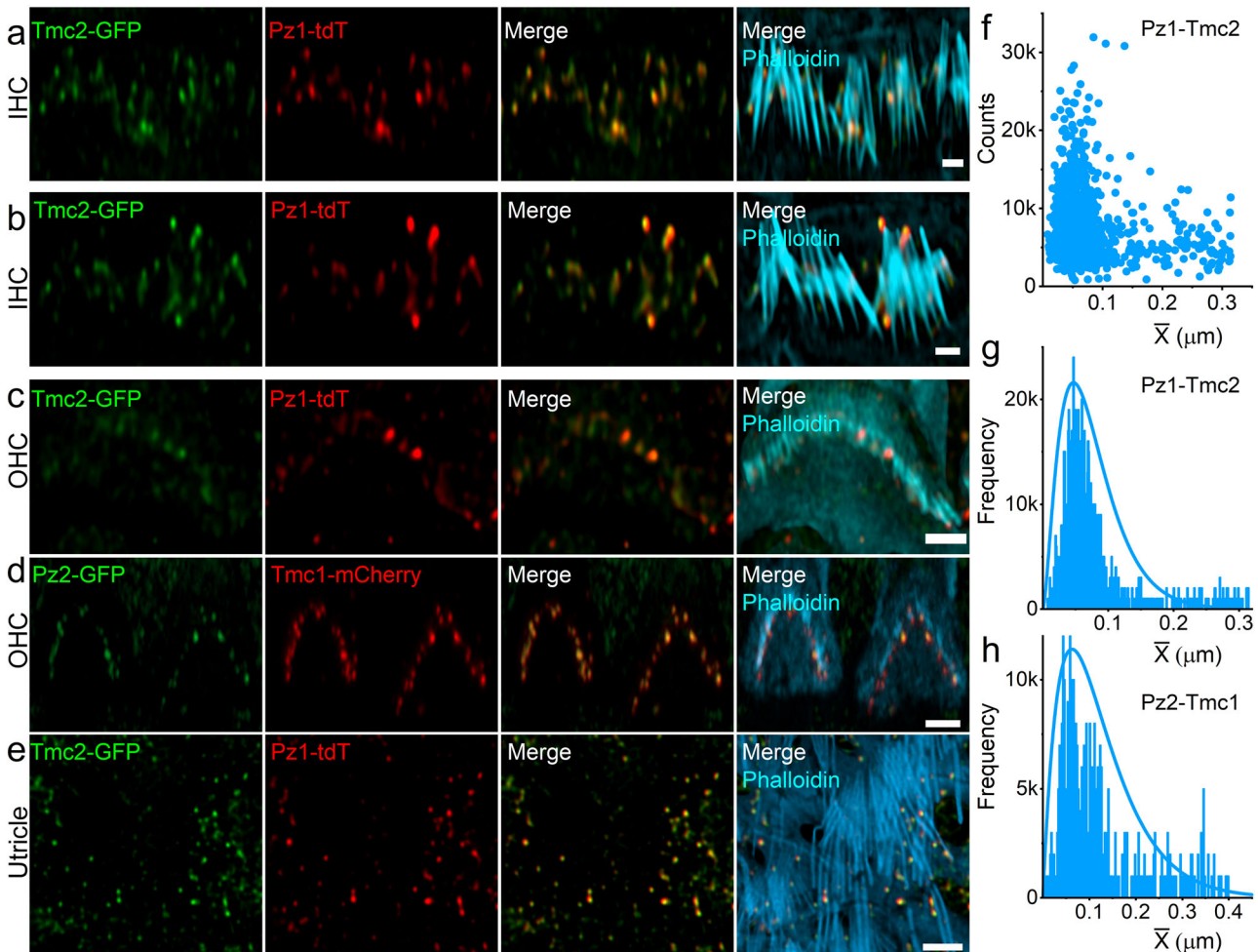

**Fig. 2 | Tmc1, Tmc2, Pz1, and Pz2 are localized in IHC and OHC stereocilia and utricular HC stereocilia bundles. a–c** Confocal fluorescence images obtained from *Tmc2-AcGFP/Pz1-tdT* transgenic mice at P10. Tmc2 (green) and Pz1 (red) labeling were strongest at stereocilia tips, and plasma membranes of cuticular plates counterstained in blue with Alexa-405-phalloidin for actin. As indicated, the Tmc2 and Pz2 expression patterns were similar in IHCs and OHCs. (Scale bar, 1 μm). **d** Expression of Tmc1 (red) and Pz2 (green) captured from an OHC of *Tmc1-mCherry:Pz2-GFP* mice. For the image shown, Pz2 labeling is seen strongly at OHC stereocilia tips and cuticular plate membrane, while Tmc1 is detected mainly at stereocilia tips. (Scale bar = 1 μm). **e** Confocal fluorescence image of utricular HC stereocilia bundles from *Tmc2-AcGFP/Pz1-tdT* transgenic mice at P10. Tmc2-GFP (green) and Pz1-tdT (red) localize predominantly to the tips of stereocilia, which were counterstained in blue with Alexa 405-phalloidin for actin. (Scale bar, 2 μm). All experiments were repeated in 5 animal samples, and at least 16 images were collected and sampled blindly by three individuals for each independent experiment (**a–d**). **f–h** Scattered plot and histogram depicting the counts and frequency plots of NND (Pz1-Tmc2) and (Pz2-Tmc1) in hair cell stereocilia. At least 5 animals in each group were sampled using 16 images. Source data (**f–h**) are provided as a Source Data file.

efficient $Ca^{2+}$ buffer ($I_{Pz1}$ $\tau_{decay}$ at −80 mV = 8 ± 4 ms ($n = 5$) and 70 mV = 97 ± 10 ms ($n = 6$)), demonstrated the $Ca^{2+}$-dependence of $I_{Pz1}$ decay. However, the displacement-response relation remained significantly unaltered using different pipette $Ca^{2+}$ buffers (S8).

We co-expressed mouse Pz1 and Pz2 channels and Tmc isoforms as proxies to test the functional significance of colocalization in HC stereocilia (Fig. 2, S4). Expression of mPz1 and mPz2 by themselves yielded distinct current profiles. Pz2 currents ($I_{Pz2}$) showed fast activation and decay kinetics (S9a). The $\tau_{decay}$ or $\tau_{inact}$ (inactivation) for maximum $I_{Pz2}$ and $I_{Pz1}$ was <10 and >10 ms, respectively. After 36–40 h post-transfection, the current density for $I_{Pz2}$ was ~2.5-fold (323 ± 78 pA/pF, $n = 17$) greater than $I_{Pz1}$ (129 ± 25 pA/pF, $n = 19$). Co-expression of *mPz1* and *mPz2* in an equivalent molar ratio (1:1) produced current ($I_{Pz1/2}$) with properties resembling a hybrid of the two-channel currents with $\tau_{decay}$ ranging from 9–24 ms (mean $\tau_{decay}$ = 17.4 ± 4.7 ms, $n = 14$). The normalized displacement-current response relationships were fitted with a two-state Boltzmann function, and half-maximum displacements ($X_{1/2}$, in μm) were estimated for $I_{Pz1}$ = 0.87 ± 0.01 ($n = 11$), $I_{Pz2}$ = 0.67 ± 0.01 ($n = 10$) and $I_{Pz1/2}$ = 0.76 ± 0.01 ($n = 12$) (S9b). There

were notable differences in MET current densities and $\tau_{decay}$ after co-expressing mPz1 with mTmc1 or mTmc2 (S9c–e). The mechanically-activated current decay for mPz1 and mTmc2 transfected N2A cells increased compared to mPz1 alone, and the $\tau_{decay}$ decreased ~4.5-fold, indicating the regulation of mPz1-mediated current by Tmc2. Another visible and consistent effect of singly and co-expressed m*Pz1, Tmc1,* and *Tmc2* was the robust amplitude of m*Pz1-Tmc1* current compared to *mPz1-Tmc2* or *mPz1* current alone. For similar transfection conditions, the total $I_{Pz1/T1}$ density (292 ± 40 pA/pF, $n = 10$) was ~2.4-fold greater relative to $I_{Pz1}$ (122 ± 18 pA/pF, $n = 10$), and $I_{Pz1/T2}$ (113 ± 22 pA/pF, $n = 10$). Co-expression of *Tmc1* or *Tmc2* with *mPz1* qualitatively shifted displacement-response relationships rightward of $I_{Pz1}$ alone, with Tmc2 exerting the most pronounced effect (S9). In contrast, the steepness of the displacement-response curves at $X_{1/2}$ remained relatively unchanged (S9f). Because the present experiments were limited by the relatively slow rise time of the mechanical stimulator and because of the coarse mechanical stimulation, quantitative analyses were not performed for the displacement-response relationship and the time constants of current activation. A cautionary note is that the

stimulus probe and features used on the N2A cells were more in keeping with those used to trigger the anomalous MET current; thus, the ensuing current may not yield regular MET current features.

## Evidence for the functional coupling of Pz and Tmc using Förster resonance energy transfer (FRET)

Motivated by Pz and Tmc colocalization in HC stereocilia and the apparent $I_{Pz}$ property modulations by Tmc1/2 co-transfection ($I_{Pz/T1/2}$), we tested additional evidence for functional coupling between the two proteins. FRET experiments were performed in OHC stereocilia isolated from mice expressing Pz2-GFP and Tmc1-mCherry that were crossbred from heterozygotes for each genotype. FRET experiments were also performed in N2A cells, co-transfected Pz1 tagged with mClover3 (mPiezo1-mClover3) and Tmc1 tagged with mRuby3 (mTmc1-mRuby3). FRET efficiencies were quantified using acceptor photobleaching (S10). Results from FRET experiments showed a FRET efficiency of ~10%. Experiments conducted using OHC stereocilia, expressing Pz2-GFP and Tmc1-mCherry, showed similar FRET efficiency (S10), suggesting that the two proteins are co-localized and functionally coupled.

## Co-transfection of Tmc1 increases the dihydrostreptomycin (DHS) block of $I_{Pz1}$ by three orders of magnitude

MET current block by aminoglycosides is a pharmacologic trademark[20,54]. Consequently, $I_{Pz}$ or current ensuing from a Pz- and Tmc association should resemble dihydrostreptomycin (DHS) effects on HC MET currents. Thus, we tested for $I_{Pz1}$ sensitivity towards DHS. DHS application on N2A cells expressing mPz1 alone inhibited the $I_{Pz1}$ in a dose-dependent manner. Results show an $IC_{50}$ of $270 \pm 135\,\mu M$ ($n = 7$), with ~65% of the maximum $I_{Pz1}$, blocked by $200\,\mu M$ neomycin ($n = 2$, S11a–c). Typically, HC MET current is at least ~20-fold more sensitive to DHS than $I_{Pz1}$ DHS-sensitivity in N2A cells (S11). Since previous results showed that *Tmc* mutations reduced HC MET current sensitivity to DHS[58,59], we tested the $I_{Pz1/T1}$ block by DHS. Pz1-Tmc1 co-transfection-current yielded an $IC_{50}$ of ~$70 \pm 6\,nM$ ($n = 7$) for DHS (S11). The ~4000-fold increased sensitivity of $I_{Pz1/T1}$ towards DHS provided additional evidence for the Pz-Tmc functional association.

## mPz1 in lipid bilayer exhibits similar single-channel conductance as MET current

HC MET single-channel conductance is ~100 pS[60,61]. We subjected the Pz1-transfected N2A-cell membrane to pressure at varying voltages, and the results showed a Pz1 unitary conductance of ~36 pS (S12), consistent with previous reports[41] but at odds with HC MET channel conductance. However, $I_{Pz}$ is modulated by regulatory and interacting proteins, and the force-response properties depend on membrane lipids[62]. Recent studies suggest that HC Tmc may exhibit lipid scramblase activity[63]. Using purified mPz1 from brain lipid extract in a lipid bilayer with a defined lipid composition (PE: PS: PC 5:3:2), we demonstrate that the unitary mPz1 current is ~10 pA at $-100\,mV$ with a single-channel conductance of ~100 pS, in keeping with the reported HC MET single-channel conductance. DHS, GsMTx4, and ruthenium blocked the mPz1-channel activity in lipid bilayers, consistent with HC MET's current pharmacology (S13).

## Knockdown of Pz by CRISPR/Cas9-mediated genome engineering

The data accrued motivated us to test *Pz* functions in vivo in the inner ear, using non-functional *Pz1* and *Pz2* gene knockin (ki) and HC-specific Cre-recombinase strategies. While other gene manipulation strategies, such as mutated-gene ki with predictable biophysical outcomes at the promoter sites, could have been adopted, we opted against such design to mitigate potential embryonic lethality because of the channels' essential roles in other tissues. Additionally, the amassed data

suggested that Pz1 and Pz2 are expressed in HCs, and the two may form heterotrimers (S2–3).

We generated two ki mice in a C57BL6/NTac genetic background using CRISPR/Cas9-mediated genome engineering (Cyagen Biosciences, Inc., Santa Clara, CA, USA). The non-functional forms of *Pz1* (*Pz1^MU*) and *Pz2* (*Pz2^MU*) were flanked by loxP sites and inserted into intron 1 of the ROSA26 locus in the two ki mice. Additionally, a control wildtype (WT) *Pz1^WT* was generated. Both the *Pz1-mutant* and *Pz2-mutant* ki (*denoted as Pz1^MU and Pz2^MU*) coding sequences contain four amino acids in the C-terminal domain (CTD), where the sequence "MFEE" was replaced with amino acids "AAAA" (Fig. 3a, b). The sequences correspond to amino acids 2493-96 for *Pz1* and 2767-2770 for *Pz2*. The selection of EE:AA residue substitutions stems from earlier reports[64] demonstrating that while mutant Pz proteins with substitutions of the AA residues can be targeted to the plasma membrane, they are non-conductive. Similar results were confirmed for MFEE:AAAA mutants transfected in N2A cells (S14). We performed analyses in silico to predict the top five most likely off-target mutations that could occur with the CRISPR approach and ensured the mice did not harbor them. Mice were also backcrossed unto C57 mouse lines for ten generations to mitigate potential off-target effects. Genotyping strategies are described (Fig. 3c, d; see Methods). We denote the mutant ki as *Pz1^MU* and *Pz2^MU*. The two mutant lines were crossed to obtain double mutant ki (*Pz1-2^MU*). A control mouse with wildtype (WT) Pz1 at the ROSA 26 locus was also generated (*Pz1^WT*).

Calretinin is encoded by the *Calb2* gene, subserving as a major $Ca^{2+}$ buffer[65,66] in HCs, whereas myosin 15 is encoded by the *Myo15* gene, showing early postnatal expression in HCs[67,68]. *Calb2* is also expressed in auditory neuron subtypes[69]. We used the two Cre lines as complementary strategies to regulate *Pz* ki mouse lines. Procedures, generation, genotyping, and validation of *Myo15*-Cre (*mc*) and *Calb2*-Cre (cc) mouse lines are described in the Methods section and Supplement 15 (S15). We first confirmed HC-specific Cre expression by crossing *mc* and *cc* with the *Ai9-tdT (Ai9(RLC-tdT;* JAX strain #00709) and *Calb2-GFP* mouse lines (S15). Additionally, we inactivated *Pz1* and *Pz2* in HCs individually, using *Pz1^MU* and *Pz2^MU* mice[44,70] crossed with *mc* and *cc* mouse lines to assess the potential differential roles of the two proteins. We will refer to the resulting lines after crosses as *mc-Pz1^MU*, *mc-Pz2^MU*, and *cc-Pz2^MU*. In contrast, *cc-Pz1^MU* mouse lines were not generated since *Calb2* and *Pz1* are located on chromosome 8. The controls used were *cc, mc,* or *Pz1^WT*, where the wildtype Pz1 channel was knockin at the ROSA 26 locus. The auditory and vestibular traits in the control mice were similar and thus used interchangeably.

Pz conditional knockout mouse models *Pz1-knockout (Pz1^KO)* cross *with the Cre-lines* were denoted as *mc-Pz1^KO*, *mc-Pz2^KO*, and *cc-Pz2^KO*. The crosses of heterozygous floxed lines with and without the Cre allele yielded offspring with a 1:2:1 ratio for the wildtype (WT), heterozygous, and homozygous floxed allele, respectively, suggesting no embryonic lethality. *Pz1^MU* and *Pz2^MU* mice appeared normal without obvious behavioral defects (rotarod test, data not shown) and body weight issues (S16a). Real-time RT-PCR analysis of wildtype and *Pz1/2^MU* cochlear samples at P10 demonstrated normal *Pz* transcript levels (data not shown). We performed immunostaining of whole-mount cochlear tissue using the Pz1 antibody and detected correctly targeted Pz1 in HC stereocilia in the *Pz1^MU* mouse samples (S16b, $n = 6$ cochleae).

## The auditory and vestibular traits of $Pz^{MU}$ and $Pz^{ko}$ mice

We analyzed auditory brainstem responses (ABR) to broadband click- and pure tones of 4, 8, 16, and 32 kHz stimuli at various sound-pressure levels (SPL). ABRs were measured at 4- and 8-weeks of age. Representative ABR traces show reduced characteristic waveform peaks and increased latencies using click sounds at 60- and 80-dB SPL (Fig. 4a). From 4 to 8 weeks, *mc-Pz1^MU* mice exhibited an ~20 to 35 dB threshold increase, while *mc-Pz2^MU* mice showed an ~40 dB threshold shift compared to age-matched control *mc* and *mc-Pz1^WT* mice for broadband

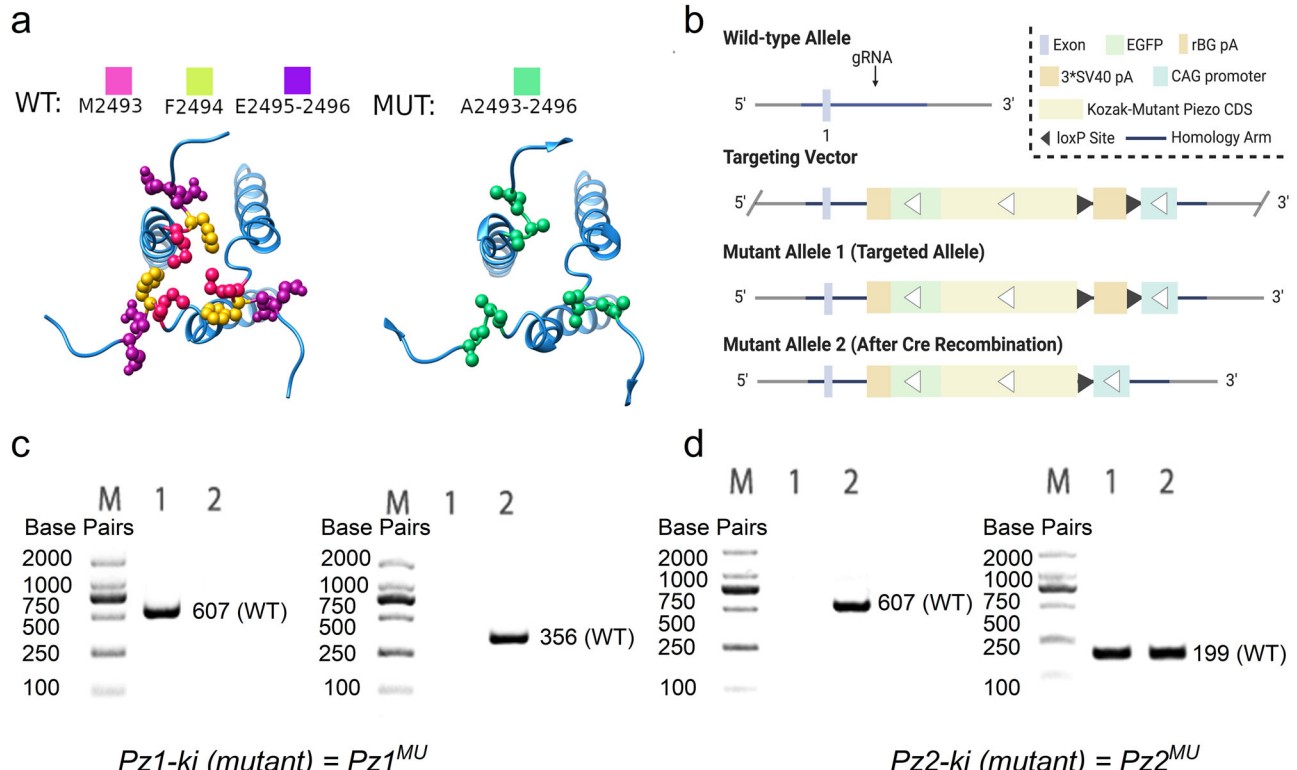

**Fig. 3 | Generation of *knockin (ki)* mouse models. a** Introduction of the mutation into the *Pz1* and *Pz2* loci by gene targeting. **a**, Cartoon of a short segment of the C-terminal domain (CTD) of Pz protein showing the sequence residues 2493–2496 in Pz1 MFEE mutated to AAAA. The corresponding conserved sequence in Pz2 was at 2767-2770, mutated to AAAA. The plasma membrane appears as a blue rectangle. Amino acids are labeled as indicated: polar, hydrophobic, acidic, and basic residues. **b** CRISPR/Cas9-mediated genome engineering was used. The *Pz* conditional overexpression *ki* mouse line features a "CAG promoter-loxP-3*SV40 pA-loxP-Kozak-Mutant *Pz1* CDS-P2A-EGFP-rBG pA" cassette inserted into intron 1 of the ROSA26 locus. The *Pz2* conditional overexpression knockin mouse line features a "CAG promoter-loxP-3*SV40 pA-loxP-Kozak-Mutant *Pz2* CDS-P2A-tdTomato-rBG pA" cassette also inserted into intron 1 of ROSA26 locus. **c, d** Tail genomic DNA was digested for *Pz1* (*Pz1-ki*) genotyping. We developed a PCR assay (35 cycles) with a pair of primers (607 bp: F:5″-AAGCACGTTTCCGACTTGAGTTG-3″ and R:5″-GGGT GAGCATGTCTTTAATCTACC-3″). The primers for the mutant were (356 bp: F:5″-TT CAGGGTCAGCTTGCCGTAG-3″ and R:5″-CTGCCGTGTGTGGACCGCATCCT-3″). The mutant *Pz2* (*Pz2-ki*) primers were: (199 bp: F:5″-ACCTCCTCGCCCTTGCTCACCAT-3″ and R:5″-TCGTGAGGGAGACAGGGGAGTTG-3″).

click (Fig. 4b) and across all tone pip stimuli (S17). Results indicated hearing dysfunction by progressively elevated ABR thresholds, with severe hearing loss greater than 70–80 dB SPL thresholds for *mc-Pz1^{MU}* and *mc-Pz2^{MU}* and profound hearing loss by 12 weeks (threshold at or above 90 dB for click sound) and concomitant threshold elevations across all tone stimuli. However, the elevated thresholds were more significant at 32 kHz than at 4–16 kHz in 4-week-old mice, and a progressive increase in ABR thresholds was observed at 8 weeks in the *mu* relative to control mice (S17). Hearing loss (threshold ≥90 dB) was recorded for the double mutant *mc-Pz1-2^{MU}* mice by 8 weeks of age (Fig. 4b and S17 also see data for 4-week-old *mc-Pz1-2^{MU}* mice). ABR thresholds for the *cc-Pz2^{MU}* were similar to those of *mc-Pz1^{MU}* and *mc-Pz2^{MU}* mice (Fig. 4, S17). In contrast, null deletion of individual *Pz1* or *Pz2* alone produced modest ABR threshold elevations, using *Myo15 and Calb2 Cre* lines and click sound as documented (Fig. 4). Attempts to generate knockouts of both isoforms, thus far, have failed. We suspect embryonic lethality despite using multiple Cre lines.

Increased ABR thresholds may suffice to capture IHC malfunction, whereas distortion product otoacoustic emissions (DPOAEs) are acoustic measurements of OHC activity. DPOAE thresholds for the *mc-Pz1^{MU}* and *mc-Pz2^{MU}* mice were elevated relative to age-matched control mice (Fig. 4d). Similar ABR and DPOAE results were obtained in the *cc-Pz2^{MU}* mouse lines. Results suggest compromised integrity of electromechanical transduction and a decline in the competence of IHC and OHC functions.

Pz channels' functional roles in vivo in vestibular end-organ HCs were examined by measuring compound action potentials, the vestibular sensory evoked potential (VsEP)[71]. VsEP originates from HCs, the vestibular nerve, and its central relay activity in response to linear acceleration pulses. Figure 4e shows VsEP waveforms for *cc* control and *cc-Pz2^{MU} and mc-Pz1-2^{MU} double* mice. The summary data show that VsEP thresholds in single *Pz1* or *Pz2* mutant knockins are significantly elevated relative to those for the age-matched controls (Fig. 4f). Compared to double mutant mice (*Pz1-2^{MU}*), relative to controls, the VsEP threshold differences were even more remarkable. However, results from a null deletion of individual *Pz1* and *Pz2* show no significant effects (Fig. 4f).

**Mechanotransduction is attenuated in *Pz^{MU}* mice**

To explain the observed hearing phenotype, we monitored the extent of altered HC transducer currents in the *Pz^{MU}* mice. The rapid uptake of the lipophilic dye FM1-43 occurs through the MET channels[17,19]. At rest, HC MET channels' $P_o$ is -0.1, so we reasoned that a nonconducting Pz subunit would further reduce the resting $P_o$ and inhibit HC FM1-43 dye uptake. We examined FM1-43 dye loading in *mc-Pz1^{MU}* and *mc-Pz2^{MU}* HCs from the apical-to-middle cochlear region, representing 4-to 10-kHz characteristic frequencies (CFs) from P12 mouse cochleae and compared them to aged-matched *mc* controls. Local perfusion of FM1-43 dye resulted in intense labeling first at the hair bundle level 1 (L1), followed by dye membrane-partitioning and diffusion across the cuticular plate (L2) and basolateral (L3) aspects of HCs. On average, chronological images from the three levels show intense labeling after 3–10 s of 10 µM FM1-43 exposure in the *mc* control and faint labeling in *mc-Pz1^{MU}* HCs (S18). Z-stack images were taken at the cuticular plate

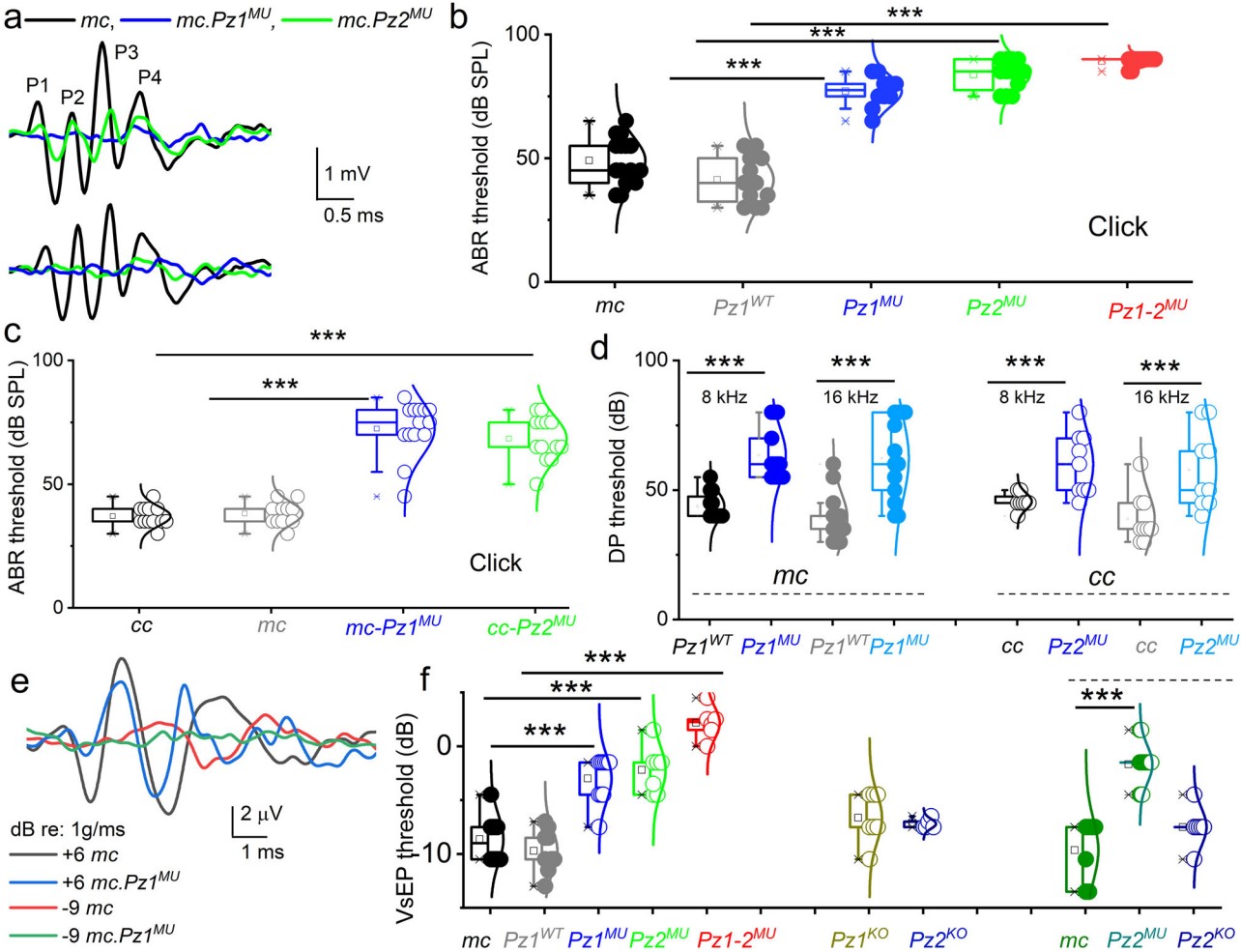

**Fig. 4 | Hearing loss and vestibular hypofunction in *Piezo mutant ((MU), Pz^{MU})*.**
**a** ABR click stimuli at 80 and 60 dB. *Myo15-Cre (mc)*, black traces, and *Pz1 (mc-Pz1^{MU},* blue) and *mc-Pz2^{MU}* (green) mice (8 weeks old). The peaks (P) of the waveforms.
**b, c** ABR thresholds to click stimuli. **b** Thresholds for mc (●), WT Pz1 ki (*mc-Pz1^{WT}*), (●), *Pz1^{MU}* (●), *Pz2^{MU}* (●) and double *Pz1-2^{MU}* (●) in 8-week-old. Data are shown as mean ± SD. Box plot, maxima, and minima are in (s), mean (□), and 99% and 1% are (X) (**b–d**, **f**). There were significant differences at the *0.05* level for *Pz1^{WT}*, one-way ANOVA, $F_{(4,56)} = (105)$ $p = 2.4 \times 10^{-25}$. *Post hoc* comparisons using the Tukey HSD test at 8-w, *Pz1^{WT}* (41 ± 10, $n = 12$), *Pz1^{MU}* (77 ± 6, $n = 10$) ($p = 3.3 \times 10^{-6}$); *Pz2^{MU}* (84 ± 6, $n = 12$) ($p = 1.4 \times 10^{-6}$); *Pz1-2^{MU}* (89 ± 2, $n = 12$) ($p = 1.9 \times 10^{-8}$) are significantly different. There were no significant differences between *mc* vs. *Pz1^{WT}* ($p = 0.076$). **c** Click thresholds for (*calb2-cre (cc(○))*), and *Pz2^{MU}* (○) at 8-w of age. One-way ANOVA, $F_{(3,46)} = (72)$ $p = 2.1 \times 10^{-17}$. *Post hoc* comparisons showed *cc* (37 ± 4, $n = 12$) vs. *Pz2^{MU}* (63 ± 9, $n = 13$) ($p = 1.9 \times 10^{-10}$). *mc* (38 ± 5, $n = 11$) vs. mc-*Pz1^{MU}* (73 ± 11, $n = 14$)

($p = 1.2 \times 10^{-7}$). There are no histological changes in cc and mc (8 and 12 weeks old). **d** DPOAE thresholds at 8 and 16 kHz in *mc, cc,* and mutant mice. At 8 kHz, threshold (dB) *mc* (44 ± 5, $n = 12$ (●)) vs. *Pz1^{MU}* (64 ± 10, $n = 13$(●)) ($p = 8.8 \times 10^{-6}$); 16 kHz, (39 ± 9, $n = 13$(●)) vs. (62 ± 16, $n = 13$(●)) ($p = 1.9 \times 10^{-4}$). For *cc* at 8 kHz, threshold (dB) *cc* (47 ± 9, $n = 9$ (○)) vs. *Pz2^{MU}* (60 ± 12, $n = 9$(○)) ($p = 7.0 \times 10^{-3}$); 16 kHz, (39 ± 10, $n = 9$(○)) vs. (58 ± 15, $n = 9$(○)) ($p = 7.4 \times 10^{-3}$). **e, f** Reduced VsEP in *Pz-ki* but not *Pz-ko* mouse lines. **e** VsEP for *cc,* and *Pz2^{MU}* mice at 8-w. Response peaks at P1 and P2. **f** VsEP thresholds. For *mc* (●), *Pz1^{WT}* (●), *Pz1^{MU}* (○), *Pz2^{MU}* (○), and *Pz1-2^{MU}* (○) mean threshold values (in dB re: 1 g/ms). *mc* = −8.6 ± 2.2(8), *mc-Pz1^{WT}* = −9.7 ± 2.0 (7) vs. *Pz2^{MU}* = −2.2 ± 2.1 (7), $p = 7.9 \times 10^{-5}$; *mc-Pz1^{WT}* vs *Pz1^{MU}* = −3.0 ± 2 (10), $p = 2.1 \times 10^{-7}$ and *Pz1-2^{MU}* = 2.2 ± 1.5 (6), $p = 1.6 \times 10^{-7}$. *cc* vs *Pz2* knockout (*Pz2^{KO}*); *Pz2^{KO}* = −7.2 ± 0.4 (5), $p = 1.2 \times 10^{-1}$. *mc* (●) vs *Pz2^{MU}* (○) and *Pz2^{KO}* (○) mice; *mc* = −9.6 ± 2.9 (7) *Pz2^{MU}* = −1.7 ± 1.7 (16), $p = 1.3 \times 10^{-4}$; *Pz^{2ko}* = −7.5 ± 1.7 (7), $p = 1.2 \times 10^{-1}$. Source data are provided as a Source Data file.

level in 5-s intervals post-dye exposure (S18). The time constants (τ) of dye loading at L2 were: 23 ± 3 s ($n = 5$) for controls; 61 ± 6 s ($n = 4$) for *mc-Pz1^{MU}*; and 58 ± 8 s ($n = 4$) for *mc-Pz2^{MU}*. For clarity, only L2 data are plotted (S18). We conclude that the expression of a non-functional Pz subunit suffices to reduce but not altogether abolish FM1-43 dye loading in IHCs and OHCs. Moreover, residual MET current is expected in the ki mutants (S23).

To examine whether the reduced FM1–43 loading in *mc-Pz^{MU}* mouse HCs matches with transducer currents and to obtain direct evidence for Pz's role in HC MET currents, we recorded IHC and OHC MET currents in the whole-cell configuration from P10-P12 cochleae (Fig. 5a, b). HCs were held at −80 mV. IHC and OHC MET current in control mice showed varied activation and adaptation kinetics, as shown previously[14,72] (Fig. 5a, b). The size of the maximum MET current was significantly smaller in *mc-Pz1^{MU}* and almost negligible in

the *mc-Pz1-2^{MU}* compared to control HCs (*mc*) (Fig. 5c). The current-displacement plots showed a notable difference in MET currents from *mc* compared to *mc-Pz1^{MU}* and *mc-Pz2^{MU}* mice. While the normalized current-displacement curve for control HCs was well-fitted with a two-state Boltzmann function, the relationships derived from the *mc-Pz1^{MU}* and *mc-Pz2^{MU}* IHCs were best matched with a three-state Boltzmann function (Fig. 5d). Similar results were obtained for OHC MET currents (Fig. 5c, f). We examined the RPI following the post-BAPTA application to disrupt the tip links. Figure 5g shows the MET current generated using sinusoidal mechanical stimuli. In Fig. 5h, i, we illustrate the remaining current after a 5-mM BAPTA solution bath application. The expanded trace, in blue, shows the features of the RPI. At P12, the maximum RPI (in pA) from IHCs was significantly smaller in mc-*Pz1^{MU}* compared to *mc* control mice (Fig. 5i).

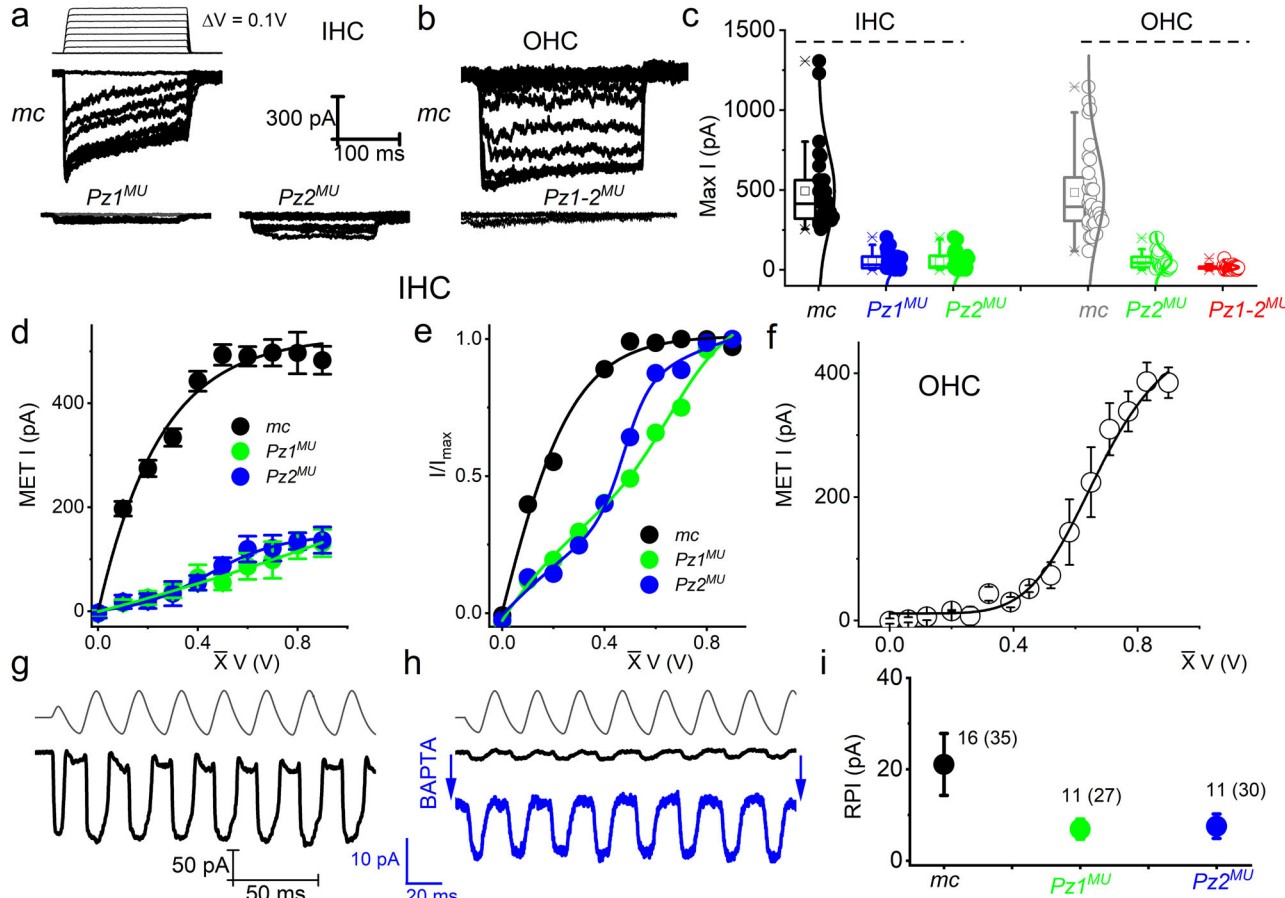

**Fig. 5 | MET currents in IHC and OHC stereocilia bundles. a** MET current in IHCs from control (*mc*), *Pz1^MU^*, and *Pz2^MU^* mice at P10 in response to fluid-jet deflection. HCs were held at −80 mV. Bundle deflection was elicited with 0.1–0.9 V pressure clamps in 0.1-V steps. For clarity, a few traces were omitted. **b** OHC traces from *mc*, *Pz1* and *Pz2* double mutant (*mc-Pz1-2^MU^*). **c** Maximum IHC and OHC current (mean ± SD), *mc*, 493 ± 255 pA (*n* = 30 IHC) from seven mice, *mc-Pz1^MU^* was 53 ± 54 pA (*n* = 30 IHCs) from nine mice, and *mc-Pz2^MU^* was 58 ± 58 pA (*n* = 30 IHCs) from nine mice. One-way ANOVA, F(2,87) = 82, *p* = 1.1 × 10⁻²⁰ *mc* vs. *mc-Pz1^MU^*, *p* = 1 × 10⁻²⁵; *mc* vs. *mc-Pz2^MU^*, *p* = 1 × 10⁻²³; *mc-Pz1^MU^* vs *mc-Pz2^MU^*, *p* = 1.0. For OHCs *mc*, 484 ± 263 pA (*n* = 32 OHCs) from eleven mice, *mc-Pz2^MU^* 58 ± 52 pA (*n* = 32 OHCs) from eight mice, and *mc-Pz1-2^MU^* (15 ± 15 pA (*n* = 29 OHCs)) from 6 mice. One-way ANOVA, F(2,90) = 85, *p* = 2.2 × 10⁻²¹ *mc* vs. *mc-Pz2^MU^*, *p* = 1 × 10⁻²⁶; *mc* vs. *mc-Pz1-2^MU^*, *p* = 1 × 10⁻³⁵; *mc-Pz2^MU^* vs *mc-Pz1-2^MU^*, *p* = 0.6. Box plot, maxima, and minima are in (−), mean (□), and 99% and 1% are (X). **d** Displacement-response relationship for *mc* (black) vs. *mc-Pz1^MU^* (green) and *mc-Pz2^MU^* (blue). **e** Normalized displacement-

response relationships fitted with a two-state Boltzmann function control (black) and a three-state function for *mc-Pz1^MU^* (green) and *mc-Pz2^MU^* (blue). **f** Displacement-response relationships fitted with a two-state Boltzmann function for OHC MET current (black) **g**–**i** Currents elicited with a 40-Hz sinusoidal deflection of P10 apical IHC *mc* with and without BAPTA. Positive-voltage evoked- inwardly directed current. BAPTA treatment reduced current by ~10-fold. Post-BAPTA currents had reversed polarity. **i** Summary data for BAPTA-mediated residual currents (RPI) in IHCs in *mc-Pz1^MU^* and *mc-Pz2^MU^* in P10 mice. The numbers of IHCs recorded are indicated in parenthesis, and the ones with measurable RPI are indicated without parenthesis. Data were obtained from at least 4 different mice in each group. Values in mean ± SD, control 21 ± 3 pA (*n* = 16); *mc-Pz1^MU^* 7 ± 1 pA (*n* = 11); *mc-Pz2^MU^* 8 ± 1 pA (*n* = 11). One-way ANOVA F(2,35) 12, *p* = 1.4 × 10⁻⁴; control vs. *mc-Pz1^MU^*, *p* = 6.3 × 10⁻⁴; control vs. *mc-Pz2^MU^*, *p* = 0.001; *mc-Pz1^MU^* vs. *mc-Pz2^MU^*, *p* = 0.98. Source data are provided as a Source Data file.

## OHC electromotility in *Pz^MU^* mutant mice was relatively intact compared to controls

DPOAE measurements, which represent the coarse assessment of cochlea-sound amplification, suggest mild OHC dysfunction in the *Pz1^MU^* and *Pz2^MU^* mouse models. We determined the roles of the Pz channels in OHCs by assessing OHC electromotility. Electromotility describes OHC functions expressed visibly as length changes in the OHC cell-body, mediated by a voltage-dependent gating-charge movement and measured as nonlinear capacitance (NLC) changes[73,74]. OHC NLC at P18-P21 in control and *Pz1^MU^* and *Pz2^MU^* mice showed substantial differences between the control and the mutant mice (S21). However, the results alone may not account for the DPOAE threshold increase[75].

## Hair cell bundle structure in *Pz1^MU^* and *Pz2^MU^* mice

Mechanoelectrical transduction appears to be a requisite for HC maturation[76], and with dwindling current magnitude in *Pz^MU^* HCs, we

expected HC morphological alterations and degeneration. We counterstained HC-stereocilia actin with phalloidin-TRITC in P12 and 4-week (P28)-old cochleae. We observed no gross changes in hair bundle structure and the characteristic three rows of OHCs and one row of IHCs in the *mc-Pz^MU^* P12 cochlea (data not shown), but by 4-week, loss of OHCs was apparent in the cochlea (Fig. 6a). We used scanning electron microscopy (SEM) for high-resolution analyses to evaluate HC and bundle morphology. Cochlear HCs of *mc-Pz^MU^* mice at P21 had normal morphology, including intact hair bundles and stereocilia linkages in both IHCs and OHCs resembling controls (Fig. 6b–d), but the loss of a few OHCs was apparent at the basal cochlea at P21 (Fig. 6d). In older *mc-Pz^MU^* mice >P42, IHC enlargement, and multiple OHC loss was evident (S20). Profound OHC degeneration at the cochlear basal and middle turns was observed at P56 and older, and other hair bundle abnormalities were noticeable, such as fused OHC stereocilia and basal-turn IHC degeneration (S20).

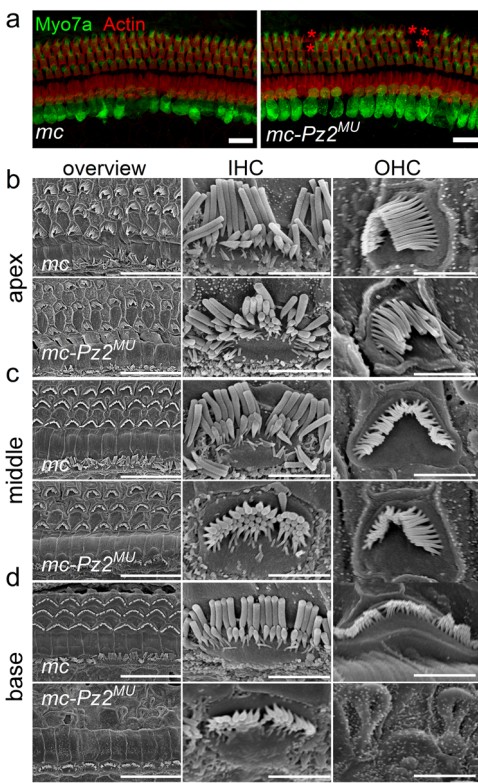

**Fig. 6 | Confocal and scanning electron images comparing cochlear HCs of control vs. mc-Pz2^MU mice. a** Left panel, Whole-mount isolated cochlea from a P28 *m-c* mouse, used as a control sample, showing one row of IHCs and three rows of OHCs, Right panel, Similar preparation from a 4-week-old *mc-Pz2^MU* cochlea. Note (* in red) indicating lost OHCs. Scale bar = 10 μm. **b–d** Overviews (left columns) showing the apex, middle, and base of 3-week-old control mice with equivalently-treated 3-week-old *mc-Pz2^MU* mice. Images show nearly identical IHCs and OHCs, although OHCs at the base are missing in certain areas in the *mc-Pz2^MU* (bottom right). Comparisons at higher magnification of IHCs (middle) and OHCs (right) are near normal in control and *mc-Pz2^MU* mice: scale bar = 10 μm, left, and 2 μm, middle and right column. Similar results were obtained from 5 control and 7 *mc-Pz2^MU* mice.

### Co-immunoprecipitation of Pz and MET complex proteins

We reasoned that if the Pz protein is an integral part of the MET protein complex, it should co-immunoprecipitate as a complex in cochlear tissue. Cochlear tissue was harvested from *Tmc2-AcGFP: Pz1-tdT* and *Tmc1-mCherry: Pz2-GFP* mice, whereas negative controls were from non-transgenic mice cochlear tissues. Results show Pz1 co-immunoprecipitated with Tmc2 and anti-GFP, as demonstrated in immunoblot using anti-tdT antibodies. Similarly, to determine whether Pz2 forms multiprotein complexes with Tmc1, we performed immunoprecipitation using anti-GFP and immunoblotting using anti-mCherry antibodies. We show that Pz1 and Tmc2, and Pz2 and Tmc1 interact in a complex (Fig. 7a). Because of limited cochlear tissue and unreliable Pz and Tmc protein antibodies, we were compelled to use the HEK 293 expression system and tagged MET apparatus proteins. Results from HEK 293 cells transiently transfected with either mCherry-Tmc1 or mCherry-Tmc2 confirmed the mCherry-tagged Tmcs pulled-down anti-mPz1 labeled Pz1, given that HEK 293 cells endogenously express Pz1 (Fig. 7b)[77]. A complement experiment using cells transfected with mCherry-Tmc1/2 and Flag-mPz1 showed a positive pull-down with anti-mCherry or anti-Flag, revealing the interaction between mTmc1/2 and mPz1. Next, similar experiments were performed using tagged proteins, including His-Tmie, Myc-Lhfp15, HA-Pcdh15, and V5-Cib2 co-expressed individually with Flag-mPz1. Co-immunoprecipitation investigations revealed that Flag-mPz1 successfully immunoprecipitated with His-Tmie, Myc-Lhfp15, HA-Pcdh15, and

V5-Cib2, suggesting the proteins form complexes (Fig. 7c–f). The original gel blots are presented in supplement Fig. 22 (S22) for disclosure and evaluation. The findings suggest that the Pz protein subunit forms complexes with the MET channel components in HEK 293 and cochlear tissue.

## Discussion

The sound and balance gateways to the brain are through MET-channel activation in HC stereocilia. However, the components forming the HC MET channel have been an enigma, the identities of which have been slowly and painstakingly forthcoming. The prevailing data suggest Tmc as the pore-forming protein[29], but it only exists as a reconstituted protein in a liposomal membrane[37]. Moreover, whereas mutations of Tmc1 in HCs may block conventional MET current, the anomalous current is spared[59]. Experiments implicating Tmc1 as a pore-forming protein in HCs did not consider Tmc's allosteric role on another channel[29]. Previous reports suggest that mutations or alterations in protein expression may provide evidence for an accessory protein rather than the sought-after pore-forming protein[78–80]. In contrast, extensive evidence already shows that the Pzs are pore-forming proteins, whereas TMC is still somewhat circumspect.

We demonstrate that IHCs and OHCs express *Pz1* and *Pz2* transcripts. Pz1 and Pz2 proteins are localized at stereocilia tips and sides of the adjacent neighboring stereocilium in IHCs, OHCs, and HCs of the vestibular end organ. The second and third OHC stereocilia rows house Pz1 and Pz2 channels. $I_{Pz}$ activation, decay kinetics, and $Ca^{2+}$ permeation are consistent with HC anomalous MET current[4]. Solitary Pz1 and Pz2 proteins are concentrated at the apical and cuticular plate membrane. The colocalization of Pz1 and Pz2 in the stereocilia and in silico analyses raised the possibility that the two subunits may form heterotrimers (S2–3)[45]. The formation of a functional heterotrimeric channel may explain the findings that knockout of one Pz subunit was insufficient to produce robust auditory and vestibular traits[48].

Previous reports showed that null deletion of *Pz1* or *Pz2* and double knockout of the two alleles using *Pax2-Cre* as an otic disc-derived cell regulator did not produce robust auditory or vestibular phenotypes[48]; thus, the results are at odds with the current findings. *Pax2* is a transcription factor identified in the developing inner ear, but the expression levels plummet to undetectable levels after E12.5 when the inner ear is not fully developed[81]. It is unknown whether post-E12.5, newly recruited otic disc cells are *Pax2*-positive. Indeed, reports have shown that *Pax2*-cre produces an incomplete deletion of genes in the inner ear[81,82], accounting perhaps for the apparent weak phenotype reported for the *Pax2-cre* double *Pz1* and *Pz2* mice[48]. However, the roles of the Pz channel in the inner ear are emerging, demonstrated by a pronounced high-frequency hearing loss in *Pz2* null mice[49].

Using ki mice harboring *Pz1/2* with reporter gene fusion proteins, we document that Pz1/2 co-localizes and forms multiprotein complexes with Tmc1/2 in vestibular and cochlear HCs. In HEK 293 cells, Pz1 forms multiprotein complexes with Tmie, Lhfp15, Pcdh15, and Cib2, all documented components of the MET complex[13], thus demonstrating an essential role for Pz in the channel complex. The MET complex likely consists of a cadre of interacting proteins, regulating the pore-forming proteins and facilitating the interactions with the tip link and cytoskeleton or plasma membrane lipids to shape the force transmission to the channel. Tmc, in particular, may enrich the local complex environment with lipid microdomains that may define the MET channel physiological traits and functional interactions to confer the current and its pharmacological sensitivity to DHS. Pz subunits form a non-specific cationic channel with biophysical and pharmacological features resembling the anomalous MET currents reported by others[11,12]. Since the Pz1-Pz1 subunit and Pz2 interphases suggest heterotrimer formation, using ki of a non-functional mutant subunit represents one of the most effective strategies to abolish channel functions, providing the data to support the roles of Pz subunits subserving a potential

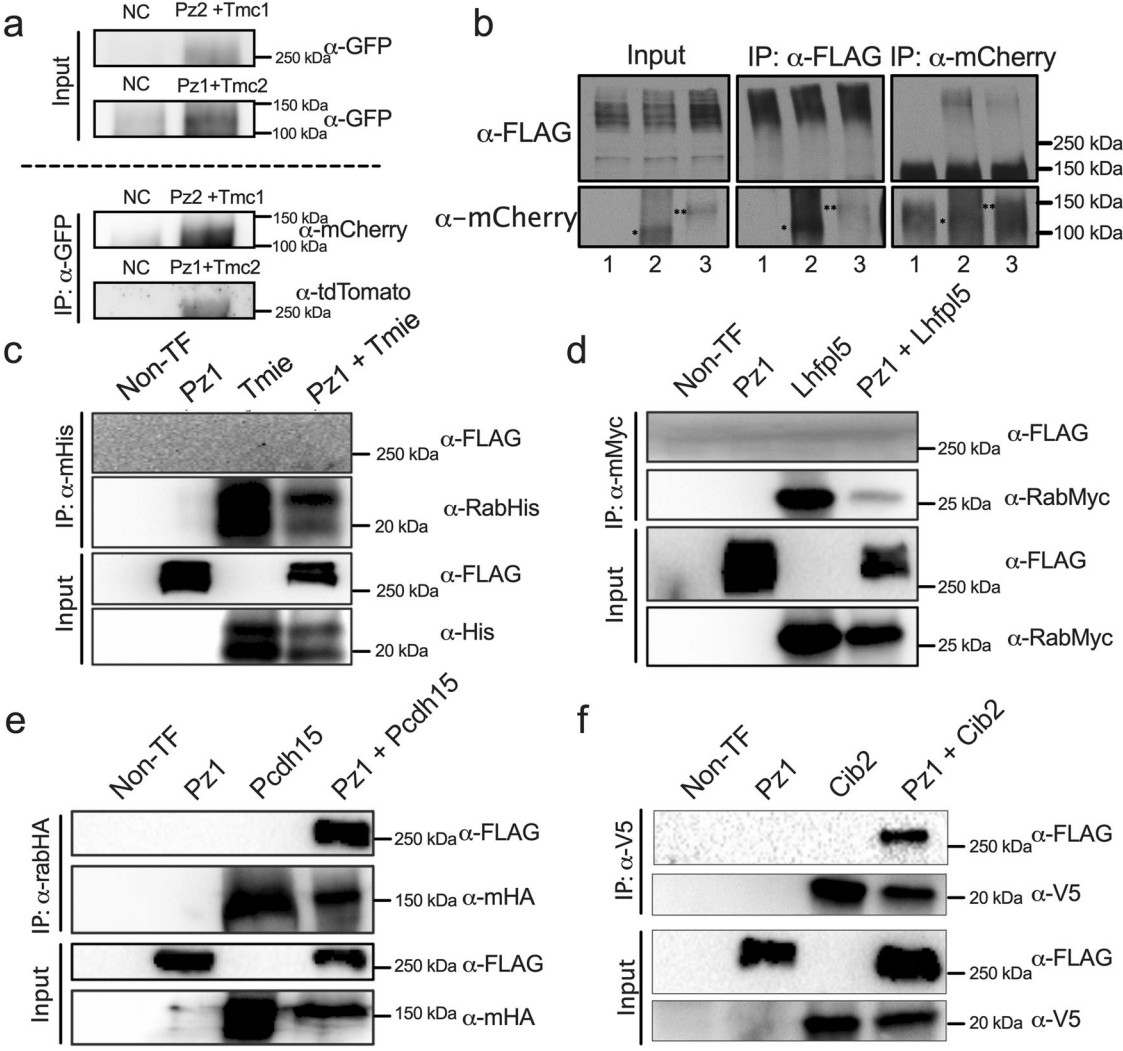

**Fig. 7 | Pz1/2 exists in a complex with Tmc1/2 in cochlear tissue and forms a protein complex with MET complex proteins. a** Representative immunoblots of GFP input and tdTomato and mCherry after IP. At least 3 independent gels were run from 20 cochleae samples. **b** Co-immunoprecipitation of FLAG-tagged Pz1 and mCherry-tagged Tmcs. Pz1-FLAG vector was transfected into HEK293 cells in combinations with mCherry alone or Tmcs-mCherry. Immunoprecipitation and western blot analysis were performed with anti-flag and anti-mCherry antibodies of a cell transfected with various combinations as listed (Lane 1: mCherry + Pz1-FLAG, lane 2: Tmc1-mCherry + Pz1-FLAG, lane 3: Tmc2-mCherry + Pz1-FLAG). Lane 1 showed a negative control. Whole-cell lysates were subjected to immunoprecipitation (IP) with a FLAG or mCherry antibody, followed by Western blotting (WB)

with FLAG or mCherry, as indicated in the figure. (* = Tmc-mCherry monomer, ** = Tmc2-mCherry monomer). Co-IP assays showing the interaction between Tmc1/2-mCherry and Pz1-FLAG (**b**), Tmie-His and Pz1-FLAG (**c**), Lhfpl5-Myc, and Pz1-FLAG (**d**), Pcdh15-HA and Pz1-FLAG (**e**) Cib2-V5 and Pz1-FLAG (**f**) and Tmc1- mCherry and Pz1-FLAG (**e**). Proteins were extracted from HEK 293 cells that transiently expressed target proteins. Protein extracts were incubated with α-FLAG and α-mCherry (**b**), α-His (**c**), α-Myc (**d**), α-HA (**e**), and α-V5 (**f**). Pull-down samples were detected using α-FLAG to identify Pz and α-mCherry to detect Tmc1 (**b**) α-His to detect Tmie (**c**), α-Myc to detect Lhfpl5 (**d**), α-HA to detect Pcdh15 (**e**), α-V5 to detect Cib2 (**f**) in western blotting. At least 5 independent batches of sHEK 293 cells were used in **b**–**f** Source data are provided as Supplement Fig. S22.

pathway for ion permeation in HC stereocilia. The strategy is authenticated since WT *Pz1* ki at the ROSA 26 locus yielded no abnormal auditory and vestibular traits and progressive hypofunction. Notwithstanding the caveats associated with expression systems, the expression of mTmc1 or 2 alone did not bear a mechanically activated current despite ~8% membrane trafficking (S6), whereas Pz1/2 did. Co-transfection of *Pz1/2* with *Tmc1/2* produced a mechanically-gated current with increased density and varied decay kinetics compared with *Pz1/2* alone. Pz1/2 and *Pz-Tmc* co-expression yielded current with MET pharmacological properties similar but not identical to HC anomalous MET current[20,55,83]. Recapitulation of HC MET current properties may require the assembly of all in vivo MET complex components. Additionally, the stimuli used to invoke mechanically activated current in N2A differs from that used to activate HC MET current; the resulting currents may not show identical features.

The null deletion of *Pz1/2* alone in HCs had a mild effect on auditory and vestibular functions[48], whereas knockin of mutant *Pz1* or *Pz2* subunits produced ABRs and VsEPs with increased thresholds and profound deafness by ~3 months accompanied by HC degeneration. Mice expressing mutant *Pz1* and *Pz2* (*mc-Pz1-2^{MU}*) were deaf by 8 weeks and exhibited severe imbalance. Moreover, IHCs and OHCs expressing *Pz* mutants have reduced FM1-43 uptake with attenuated HC transduction currents, substantiating the assertion that Pz is necessary for HC MET current. The residual current and remnants of FM1-43 uptake likely resulted from endogenous WT Pz1/2 channels (S23). Similar results using two distinct HC Cre lines (*Myo15* and *Calb2*) support the conclusion that Pz subunits are essential components of the MET apparatus. HC Pz and Tmc proteins form multiprotein complexes, and the results were confirmed in HEK 293 and N2A cells.

Evidence for HC *Pz2* transcripts and protein expression was previously shown[48,49], but detection was not reported for *Pz1*[48]. A more sensitive hybridization strategy in situ, smFISH[84], allowed for *Pz1* and *Pz2* RNA detection and quantification in HCs, while high-resolution microscopy enhanced by protein-fluorophore fusion expression revealed Pz and Tmc proteins localized at the stereocilia tips and sides in propinquity (Figs. 1–2). Given the close localization of Pz1/2 in HC stereocilia, the predicted structural and functional interaction between the two subunits in HCs, N2A cells, and the trimeric structure of the Pz subunits, we sought to attenuate Pz channel function in vivo using non-functional *Pz1* and *Pz2* mutants. The auditory and vestibular phenotypes of the *Pz1^MU*, *Pz2^MU*, and double *Pz1-2^MU* mice are consistent with their roles in HC MET function. Considering a stochastic assembly of channel subunits and assuming functional hetero- and homomeric channels and equal levels of subunit expression, ~30% of HCs may carry functional WT Pz subunits in the single mutant allele ki HCs, and the functional-channel numbers decrease to ~12% in the *Pz1-2^MU* cochlear HCs (S23). Moreover, the analysis may explain the residual FM1-43 uptake (S18), hearing, and vestibular traits in single *Pz^MU* mice in contrast to the early and severe traits in double *Pz1-2^MU* mice (Fig. 4).

Tmc is an integral element of the MET complex in mammalian HCs and may serve as an ion channel[29,36]. Cell lines expressing Tmc provide evidence for Pz1 interacting with MET complex proteins, including Tmie, Lhfp15, Pcdh15, and Cib2, indicating that the Pz subunits are part of the MET complex in both cochlear tissue and a reconstituted system. Tmc may be an essential allosteric regulator for Pz-channel functions in mammalian HC. Among the prevailing evidence that Tmc serves as the pore-forming subunits for MET channels is a set of elegant experiments in which specific Tmc1 residues were mutated into cysteine, and cysteine-modification reagents were able to alter MET currents[29]. In a scenario where the Pz-channel complex was intact except for the mutated Tmc1, it is conceivable to use cysteine modification to restore the MET complex function. Suppose Tmc is an allosteric anchor for force transmission through lipid interaction with Pz, and Tmc is required for Pz engagement in the HC stereocilia setting. In that case, results can be envisioned from the previous report.

In contrast, an improbable setup for two parallel ion permeation pathways, one for the Tmcs and the other for the Pzs might be proposed and tested. However, the simplest explanation for the current findings is that the Pz channel is the pore-protein, and a vital allosteric regulatory protein, such as Tmc and other recognized binding partners, sculpts the MET's current properties. From the findings, Pz-Tmc current sensitivity to aminoglycosides also offers persuasive evidence for the importance of Tmc in the function of Pz in HCs. The findings provide a compelling alternative for the MET channel identity, Pz protein forming a pore and complexing with documented components of the MET channel elements to confer its gateway function for sound-balance-brain communication.

Limitations of the current findings that the Pz channel constitutes a component of the MET complex are as follows. Suppose Tmc is an allosteric regulator of Pz; the mechanisms underpinning altered $Ca^{2+}$ selectivity and DHS channel-pore block by allosteric modification and the implicit suggestion that the methanethiosulfonate (MTS) reagents' effect on Tmc function is state-dependent[29] remain to be fully explained. In the current study, only *mc-Pz1^WT* was used as one of the control mice. We expect *mc-Pz2^WT* mice to have similar auditory and vestibular phenotypes; albeit unlikely, the mice may exhibit an unanticipated trait. Although a few human gain-of-function *Pz1* and *2* mutations have been reported[85], none have reported altered auditory or vestibular hypofunction to date. As more studies in the fledgling field of Pz-channelopathy emerge, future findings may clarify such ambiguity. Finally, despite the potential for embryonic lethality, attempts to generate a biophysically predictable Pz mutant model would be gratifying to interrogate the MET current and auditory and vestibular functions. For an emerging field of Pz channel functions, impending discoveries are anticipated to enhance and elucidate any perceived uncertainties raised in the current findings.

## Methods

### Animal models

**Generation of genetically modified mouse strains.** All animal experiments were performed under the University of Nevada Reno Institutional Animal Care and Use Committee guidelines under protocol number 20-08-1069-1 authorized for the Yamoah laboratory. Equal numbers of males and females were used in the experiments. In cases where odd numbers are reported, females outnumbered male samples. Data from males and females were segregated. However, we did not detect statistical differences in the auditory and vestibular traits. Thus, we combined data from both sexes. Euthanasia was conducted using high-dose pentobarbital as authorized by the protocol. To generate the *Myo15-Cre* mouse line, a "P2A-Cre-T2A-E2-Crimson" cassette was targeted to replace the TGA stop codon in exon 66 of the *Myo15* gene. The *Calb2-Cre* mouse line was created by inserting a "P2A-Cre-IRES-TagBFP2" cassette upstream of the TAA stop codon in exon 11 of the *Calb2* gene.

For Piezo1 genotyping, we developed a PCR assay (35 cycles) with a pair of primers (607 bp: F:5″-AAGCACGTTTCCGACTTGAGTTG-3″ and R:5″-GGGTGAGCATGTCTTTAATCTACC-3″). The primers for the mutant were (356 bp: F:5″-TTCAGGGTCAGCTTGCCGTAG-3″ and R:5″-CTGCCGTGTGTGGACCGCATCCT-3″) (Fig. 4c). The mutant Peizo2 primers were: (199 bp: F:5″- ACCTCCTCGCCCTTGCTCACCAT-3″ and R:5″-TCGTGAGGGAGACAGGGGAGTTG-3″) (Fig. 4d). *Myo15-cre* PCR assay was performed with a pair of primers for wildtype, wt: (435 bp: F:5″-GGGCTTGGCAGTGGTAGTGGTAT-3″ and R:5″-GTCACTTGGTCTGGAGAGGCTG-3″), and homozygote, homo: 729 bp: F:5″-CCCAGGAAGCAGTTTAGCAGTG-3″ and R:5″-TTTGGTGTACGGTCAGTAAATTGGAC-3″). *Calb2-cre* PCR assay was performed with a pair of primers for wildtype, wt: (525 bp: F:5″-GTTGATAGGAAGGTCCATTCGGT-3″ and R:5″-CAGAAGCCTAAATCATACAGCGAAG-3″), and homozygote, homo: 271 bp: F:5″- ACTTAAACCCACTCTCACCTCTTT-3″ and R:5″-TACGGTCAGTAAATTGGACACCTT-3″)) (Supplement Fig. 7). Other mouse lines were purchased from Jackson Labs (Bar Harbor, Maine, USA). Piezo1-tdTomato https://www.jax.org/strain/029214, B6;129-Piezo1tm1.1Apat/J, strain #:029214; Piezo2-GFP https://www.jax.org/strain/027719, B6(SJL)-Piezo2tm1.1(cre)Apat/J, strain #:027719; Tmc1-mCherry https://www.jax.org/strain/028392, B6.Cg-Tg(Tmc1/mCherry)2Ajg/J, strain #:028392; and Tmc2-GFP https://www.jax.org/strain/028517, B6.Cg-Tg(Tmc2/AcGFP)3Ajg/J, strain #:02851; and Ai0-tdTomato https://www.jax.org/strain/007909, B6.Cg-Gt(ROSA)26Sor^tm9(CAG-tdTomato)Hze/J. Genotype followed primers recommended by Jackson Labs.

### Immunofluorescence

Cochleae were dissected from the temporal bone and fixed with 4% paraformaldehyde in phosphate-buffered saline (PBS) for 2 h on the rocker at 4 °C. The samples were decalcified in 0.25 M ethylenediaminetetraacetic acid (EDTA) solution for up to 3 days (duration was age-dependent) with daily solution changes. Decalcified cochleae were washed in PBS and micro-dissected to the apex, middle, and base. The utricle was dissected. For the cochlea, the tectorial membrane and Reisner's membrane were removed. Each sample was permeabilized in 0.05% Triton X-100 in PBS for 10 min and then incubated for 60 min in a blocking solution containing 10% bovine serum albumin (BSA) or 10% goat serum in PBS containing 0.1% Triton X-100. The samples were incubated with the primary antibodies overnight at 4 °C. The rinsed tissues were incubated (2 h; RT) in a fluorescent dye-conjugated secondary antibody. The following primary and secondary antibodies were used: Anti-GFP antibodies (Abcam), anti-mCherry antibodies (Abcam and Novous), anti-tdTomato (MyBioSouce.com), anti-Myo7a (Proteus), anti-Piezo1 (Novus), Alexa Fluor™ 488 goat anti-rabbit, Alexa Fluor™ 647 goat anti-rabbit, Alexa Fluor™ 647 donkey anti-goat

IgG, Alexa Fluor™ 488 goat anti-mouse IgG1, Alexa Fluor™ 568 goat anti-mouse IgG2a, Alexa Fluor™ 568 goat anti-chicken (Invitrogen), and Phalloidin (Abcam and Sigma-Aldrich). Images were captured with Leica SP8 confocal microscopes.

## Single-molecule fluorescence in situ hybridization with RNAscope

Mice were anesthetized and transcranially perfused with Diethyl Pyrocarbonate (DEPC)-treated PBS at P10. The temporal bones were isolated and fixed with 4% PFA in DEPCtreated PBS for 24 h at 4 °C to preserve RNA. The samples were decalcified in bone decalcification buffer (ACD) overnight at 4 °C. Decalcified cochleae were washed in DEPC-PBS three times. The cochlear samples were sequentially dehydrated in 10, 20, and 30% sucrose solution at 4 °C for 1 h, 2 h, and overnight, respectively, then embedded in OCT for cryosection. Samples were cryo-sectioned to a thickness of 10 μm, placed onto Superfrost slides, and stored at −80 °C until processed. Probe hybridization was performed according to the manufacturer's instructions (Advanced Cell Diagnostics, ACD). Sections were immersed in pre-chilled 4% PFA for 15 min at 4 °C. Sections were then dehydrated at RT in 50%, 70%, and twice in 100% ethanol for 5 min each and allowed to dry for 1–2 min. Fixation and dehydration were followed by protease digestion, using protease for 30 min at RT. Sections were then incubated at 40 °C with the following solutions: (1) target probe in hybridization buffer A for 3 h; (2) preamplifier in hybridization buffer B for 30 min; (3) amplifier in hybridization buffer B at 40 °C for 15 min; and (4) label probe in hybridization buffer C for 15 min. After each hybridization step, slides were washed with washing buffer three times at RT. The label probes were conjugated to Alexa Fluor 488 and 594 for fluorescent detection. Probes and positive and blank negative controls obtained from ACD are shown (Supplement Fig. 1 (S1)). Sequences of the target probes, preamplifier, amplifier, and label probe are proprietary. Detailed information about the probe sequences can be obtained by signing a non-disclosure agreement provided by the manufacturer. Incubation in DAPI solution for 15 s at RT was performed to label cell nuclei. Slides were then mounted in Fluoromount-G and sealed under a coverslip. Images were captured with a Leica SP8 confocal microscope. Dots in each fluorescent-positive cell were counted and scored as described.

## Auditory brainstem responses (ABR) and distortion product otoacoustic emissions (DPOAEs)

Mice were anesthetized using ketamine (40 mg/kg) and xylazine (10 mg/kg). Animals were then placed in a sound-attenuated chamber. Body temperature was maintained at $36.5 \pm 0.5\,°C$ using a homeothermic blanket control unit (Harvard Apparatus) and rectal probe feedback. Ground and recording electrodes were placed subcutaneously in the scalp, and a calibrated transducer (Tucker Davis) was placed in the right pinna. At a rate of 20 Hz, with intensity from 0 to 90 dB sound pressure level (SPL; rms for click stimuli) in 5 dB increments, 0.1 ms broadband clicks, and 3 ms pure tone pips at 4, 8, 16, and 32 kHz were presented. The ABR activity was extracted from 128 to 1024 stimuli. The hearing threshold was the minimum sound intensity eliciting a typical ABR waveform. For DPOAE experiments, mice were anesthetized with avertin for ABR measurements. After visual inspection to ensure a healthy external and middle ear, the mice were placed in a sound-attenuated chamber, and a dual acoustic probe/microphone assembly (Etymotic Research) was placed in the ear. Primary tones with an f2/f1 ratio of 1.25 were presented at equal sound pressure levels, and tones were routed to independent transducers and allowed to mix acoustically in the ear canal. A digital signal processor sampled and synchronously averaged the SPL for geometric mean frequencies <20.1 kHz. Above 20.1 kHz, the sampling input was automatically switched to the dynamic signal analyzer for frequency analysis to avoid DSP aliasing artifacts >22.1 kHz. Cubic (2f1–f2) DPOAE levels and corresponding noise floor levels were calculated. DP thresholds were generated for each stimulus level and were evaluated relative to the noise floor (DPOAE > 5 dB above the noise floor).

## Vestibular sensory evoked potential measurement (VsEP)

Mice were anesthetized using ketamine (40 mg/kg) and xylazine (10 mg/kg), and VsEP measurements were recorded as described previously[86]. The skull was stabilized with a head mount. Stimuli were linear acceleration pulses of 2 ms duration, nine pulses per second, presented in standard and inverted directions. Normal polarity was defined as the upward displacement of the shaker platform while inverted in the downward platform displacement. Stimulus amplitude was measured in jerk (da/dt, i.e., g/ms, where $1.0\,g = 9.8\,m/s^2$ and $1.0\,g/ms = 9.8\,μm/ms$, using a calibrated accelerometer attached to the shaker platform. Stimulus amplitude ranged from −18 to +6 dB re: 1.0 g/ms, adjusted in 3-dB steps. Two-channel signal averaging was used to resolve vestibular responses from background electroencephalographic activity. The electrophysiological activity was amplified (200,000X), filtered (300 to 3000 Hz), and VsEPs were recorded to normal and inverted stimulus polarities (1024 points, 10-μs per point, 128 responses per averaged waveform). Three response parameters were quantified: threshold, peak latencies, and peak-to-peak amplitudes. The threshold is measured in dB re 1.0 g/ms and is defined as the stimulus amplitude midway between that which produced a discernible VsEP and that which failed to produce a response. The threshold measures the sensitivity of the gravity receptor end organs, utricle, and saccule. Thresholds, latencies, and amplitudes are averaged for each group and analyzed using variance (one-way ANOVA).

## Cell culture and plasmid vector transfection

Neuro2A cells (N2A; ATCC, cat no. CCL-131) were grown in Eagle's Minimum Essential Medium (EMEM) containing 10% fetal bovine serum and 50 units/ml penicillin at 37 °C with 5% $CO_2$. Cells were plated onto 35-mm dishes or 15-mm round glass coverslips placed in 4-well plates and transfected using lipofectamine 3000 (Invitrogen) according to the manufacturer's instruction. For the Pz overexpression experiment, only 1 μg/ml of mPz1-IRES-GFP or vector was transfected, and cells were recorded 24–48 h later. The initial mPz1 and mPz2 clones were a generous gift from Dr. A. Patapoutian.

## In vitro and ex vivo fluorescence resonance energy transfer (FRET) experiments

Mouse Pz1 tagged with mClover3 (mPz1-mClover3) and mTmc1 tagged with mRuby3 (mTmc1-mRuby3) were transiently transfected into N2A cells using Lipofectamine 3000 (Invitrogen) following the manufacturer's instructions. For further immunofluorescence microscopy and FRET experiments, cells were passaged and seeded on 35-mm glass bottom dishes for imaging 2–3 days after transfection. Before FRET imaging, the cell culture media in the dishes was replaced with Tyrode's solution containing (in mM) 144 NaCl, 1 $MgCl_2$, 4 KCl, 2 $CaCl_2$, 10 D-Glucose, 0.33 $NaH_2PO_4$, 10 Hepes with pH 7.4. A Zeiss LSM 700 microscope was used for the imaging. The Pz1-mClover3 (donor) and mTmc1-mRuby3 (acceptor) co-transfected N2A cells were excited by 488 nm and 555 nm laser, respectively. For photobleaching of the acceptor signal, a specific surface membrane region (region of interest (ROI)) was chosen and photobleached with a 555 nm laser, and simultaneous donor emission signals excited with a 488-nm laser were acquired using time-lapse mode till steady state recording. The intensity increase of the donor emission was calculated by subtracting the background (averaging of 3 data points) from the steady-state averaged data (3 data points). The FRET efficiency (E) was calculated based on the donor intensity pre ($D_{pre}$) and post ($D_{post}$) acceptor photobleaching: $E = (D_{post}-D_{pre})/D_{post}$. To minimize the photobleaching

effects of the donor on the FRET quantification, we used only donor cells for acceptor bleaching control, and a linear equation fitted the donor bleaching time course to obtain the slope, which was used for correction of the time course of the donor signal in the co-expression experiments.

## FM1-43 dye loading

A stock solution, 10 mM, was prepared in PBS. A new working solution was prepared. Cochlear samples were incubated in 10 μM of FM1-43 for 5 s, washed with PBS, and observed under the Leica SP8 confocal microscope as described[19].

## Scanning electron microscopy (SEM)

Mice were perfused with 4% PFA in 1x PBS, the inner ears were isolated, and the stapes footplate was removed. The ear was flushed with, then fixed overnight in 4% paraformaldehyde (PFA) and 2.5% glutaraldehyde in 1x PBS. After washing in ddH$_2$O for 3X in 1 h, the samples were post-fixed with 1% osmium tetroxide for approximately 1 h. The sample was washed in PBS before decalcifying for 3–4 days in 0.25 M EDTA at 4 °C with daily solution changes. The cochleae were microdissected, the tectorial membranes removed, and gradually dehydrated using 30%, 50%, 70%, 80%, 90%, 100% ethanol, 2:1 ethanol/ hexamethyldisilazane (HMDS) (Thermo Scientific #A15139 AE), 1:2 ethanol/HMDS, and finally 100% HMDS. Samples were transferred to an open well plate in HMDS and allowed to air dry overnight in a fume hood. Samples were then mounted on aluminum stubs (Ted Pella #16111) using double-sided carbon tape (EMS #77817-12) and stored in a specimen mount holder (EMS #76510) and sealed in a desiccator until sputter-coated with Au/Pd (Emitech Sputter Coater K550). Samples were viewed, and images were captured under SEM (Hitachi S-4800- University of Iowa Central Microscopy Research Facilities). An accelerating voltage of 5-kV and 11-kV was used (Supplement Fig. 10 (S10)). Images were compiled using CorelDRAW X7 graphic software.

## Co-immunoprecipitation

We homogenized and sonicated cochlear tissues in Pierce immuno-precipitation lysis buffer and incubated the lysates with GFP-antibody at 4 °C overnight. The next day, magnetic beads were prepared according to the manufacturer's instructions (ThermoFisher). We then incubated the sample/antibody with Dynabeads Protein G for 30 min at RT to allow the antigen-Ab complex to bind to the magnetic bead. A magnetic strip was used to remove the supernatant. The beads were then washed 3 times with a washing buffer. An elution buffer was used to elute the antigen of interest. Samples were heated for 10 min at 70 °C before immunoblotting. The primary antibodies used for immunoblots were anti-mCherry (Abcam, ab183628), 1:1000, anti-GFP (Abcam, ab6556), 1:1000, and anti-tdTomato (OriGene, TA183027), 1:1000.

Plasmids (m*Pcdh15, mTmie, mLfhpl5, mCib2,* and m*Tmc1*) were purchased from Gene copoeia. HEK 293 cells (ATCC, cat no. CRL-1573) and lipofectamine were used for plasmid overexpression. IP lysis buffer (Thermo Scientific, cat no. 87787) and dynabeads Protein G immunoprecipitation kit (Invitrogen, cat no. 10007D) were used for immunoprecipitation. 3.48x10$^6$ cells/well were seeded in a 100 mm dish 1-day before the transformation. Transfected cells were harvested 48 h after transformation and then lysated with an IP lysis buffer. The lysate sample was centrifuged, and the supernatant from the lysate sample was used in the reaction. Antibody and dynabeads G were incubated for 10 min at RT to make antibodies conjugate to beads. Antibodies conjugated beads were washed twice and incubated with the supernatant from cell lysis for 20 min at RT. The sample was washed three times and transferred to the new tube. The transferred sample was washed once, and all the washed solutions were removed. The elution buffer was incubated with the sample for 5 min at RT and boiled with 5x loading dye at 70 °C for 10 min. The boiled sample was

electrophoresed in 10% Mini-Protean TGX precast protein gels (Bio-Rad, cat no. 4561033). The sample was transferred to PDVF and then incubated with 5% skim milk for 1 h. The primary antibody reacted with the sample transferred membrane overnight at 4 °C and was washed 3 times with 1% TBST. The secondary antibody was incubated with the membrane for 2 h and was washed 3 times. The membrane was incubated with Clarity western ECL substrate (Bio-rad, cat no. 1705060), and the signal was detected with the ChemiDoc MP imaging system (Bio-Rad). Plasmids from Genecopoeia were as follows: mouse Pdch15 (EX-Mm28779-M07-GS), mouse Tmie (EX-Mm05449-M77-GS), mouse Lfhpl5 (EX-Mm30626-M09), mouse Tmc1 (EX-Mm02159-M56) and mouse Cib2 (CS-Mm07673-M02-01J). Antibodies were as follows: anti-HA (Abcam, cat no. ab9110, ab18181), anti-His (Abcam, cat no. ab18184, ab9108), anti-Myc (Abcam, cat no. ab32, ab9106), anti-mCherry (Abcam, cat no. ab183628, ab125096) and anti-Flag-HRP (Abcam, ab49763).

## Electrophysiology

Patch-clamp experiments were performed in a standard whole-cell configuration using an Axopatch 200B amplifier (Axon Instruments). Patch pipettes had a resistance of 2–3 MΩ when filled with an internal solution consisting of (in mM) 133 CsCl, 10 HEPES, 5 EGTA, 1 CaCl$_2$, 1 MgCl$_2$, 4 MgATP, and 0.4 Na$_2$GTP (pH adjusted to 7.3 with CsOH). The extracellular solution consisted of (in mM) 130 NaCl, 3 KCl, 1 MgCl$_2$, 10 HEPES, 2.5 CaCl$_2$, and 10 glucose (pH was adjusted to 7.3 using NaOH). Currents were sampled at 20 kHz and filtered at 2 kHz. Voltages were not corrected for a liquid junction potential. Leak currents before mechanical stimulations were subtracted offline from the current traces. 10 mM dihydrostreptomycin and neomycin stock solution were prepared in water. All electrophysiological experiments were performed at RT (21–22 °C). Reagents were obtained from Sigma Aldrich unless otherwise specified.

## Mechanical stimulation

Mechanical stimulation was achieved using a fire-polished glass pipette (tip diameter 1-μm) at 60° to the recorded cell. Downward movement of the probe toward the cell was driven by a Clampex-controlled piezo-electric crystal micro stage (E660 LVPZT Controller/Amplifier; Physik Instruments). The probe was typically positioned close to the cell body without visible membrane deformation. Mechanical displacement was applied with a fire-polished blunt pipette driven by a piezo-electric device. To assess the mechanical sensitivity of a cell, a series of mechanical steps in -140 nm increments were applied every 10 to 20 s, which allowed for full recovery of mechanosensitive currents. Inward MA currents were recorded at a holding potential of −80 mV. For I-V relationship recordings, voltage steps were applied 750 ms before the mechanical stimulation from a holding potential of −60 mV.

For mechanical stimulation of hair bundles, we employed step-wise and sinewave stimulation of OHCs and IHCs using a piezo-driven fluid-jet hair bundle mechanical stimulator on P10-P12 cochleae and recorded IHC and OHC MET currents in the whole-cell configuration. We represented the bundle displacement in the form of applied piezo-driven voltage. Bundle displacement was not calibrated for each cell because of variations in stimulating probe positions relative to the stimulated hair bundle.

## Estimating the pore size

We followed the approach used for the MET channel in hair cells to estimate the Piezo1 pore size from the inner face and amines radius. Plots were fit empirically to

$$\frac{I_x}{I_{Cs}} = A\left(1 - a/r^2\right)$$

where $a$ is the radius of the amines, $r$ is the radius of the channel, and $A$ is a scaling factor. Internal solutions were made with monovalent amines of different sizes, with size estimated as previously described from CPK models[86,87].

The external face pore size data were fit to

$$I_x/I_{Na} = (A(1 - a/r)^2/a)$$

where $I_x/I_{Na}$ is the current ratio, $a$ is the radius of the amine compound, $r$ is the radius of the channel, and $A$ is a scaling factor[56]. The external solution used the same approach to substitute $Na^+$ with amines of different sizes. Amines were purchased from Sigma Aldrich.

### Single-channel recordings and analysis

Single channel gating properties were recorded using a Planar Lipid Bilayer Workstation (Warner Instruments LLC, Hamden, CT). A bilayer lipid membrane (BLM) was formed across a 200-μm diameter aperture drilled within a polysulfone cup within a BLM chamber. The composition of the BLM was either a synthetic lipid blend composed of phosphatidylethanolamine/phosphatidylserine/phosphatidylcholine (PE/PS/PC) (5:3:2 w/w; Avanti Polar Lipids, Alabaster, AL) or a brain total lipid extract dissolved in 30 mg/ml n-decane (Acros, New Jersey). The BLM formed two electrically isolated liquid-filled sides of the chamber (0.8 ml each), defined as *cis* and *trans* sides, respectively. Voltage input was applied to the *trans* side, whereas the *cis* was grounded. Piezo enriched (purified) protein preparation, ~50–70 mg/ml, and test reagents were always added to the *cis* side solution (500 mM KCl, 10 mM MOPS, pH 7.4) and stirred constantly. An initial gradient of 5:2 $K^+$ *cis* vs. *trans* was used to promote the incorporation of channel protein into the BLM. Once the channel was incorporated, the *trans* chamber was immediately perfused with 500 mM KCl to establish symmetrical ionic conditions across *cis/trans* to prevent additional channel fusions. The amplified current signals were recorded at the indicated holding potentials, filtered at 800 Hz (low-pass Bessel filter 8-pole; Warner Instruments), and digitized at a sampling rate of 10 kHz (Digidata 1440; Molecular Devices, Sunnyvale, CA). Channel data was acquired with pClamp 10 software (Molecular Devices, Sunnyvale, CA). Data analysis and plots were made with OriginPro 2018b and Corel Draw 2020.

### Sequence and structural evidence for Pz1 and Pz2 heterodimers

To assess whether Pz1 and Pz2 could form heterodimers, we explored the concept of conservation of salt bridges as matching positive and negative charges is widely used to design drug and protein ligands. Using the Pz1 structure (PDB: 6BPZ), we selected one chain (see S2) (a) of the Pz1 structure and expanded our selection to include any amino acid on the remaining two chains (b, c) within 5 Å. We then kept this selection on the two chains (b, c) and expanded this selection to any amino acid on the original chain (A) by 5 Å to back-select amino acids on chain (A) to obtain all inter-chain interacting amino acids in the Pz1 trimer. We then selected the basic and acidic amino acids and mapped these onto the Pz1 mouse structure (S2–3). We then calculated a local sequence alignment between mouse-Pz1, human-Pz1, mouse-Pz2, and human-Pz2. The local alignment included the trimer interacting amino acids and surrounding amino acids to align the interacting amino acids properly. We determined the conservation of the interacting amino acids, which reports a 74.59% conservation and an 83.6% amino acid chemical property conservation (S3). The sequence on Pz2 that matched the Pz1 basic and acid salt bridges was also mapped onto the mouse-Pz2 structure (PDB: 6KG7). Due to this very high conservation, the chains could be interchanged. We verify that the conserved ionic charges match in 3D coordinates and the location of the non-conserved charges. We replaced Chain B in Pz1 with Chain B from the Pz2 structure using the matchmaker (UCSF Chimera) to make a 2:1 Pz1:Piezo2 structure. We repeated the

procedure to replace chain C of Pz1 with chain C of Pz2 to build a 1:2 Pz1:Pz2 model. Visual inspection of these models shows the pore domain contains the most conserved charged residues while the cap domain contains the most non-conserved charged residues (S2). In addition, the 3D positions of the acidic and basic residues have high conservation, suggesting that heterotrimer structures are structurally plausible.

### Statistics and reproducibility

Where appropriate, pooled data are presented as means ± SD. Significant differences between groups were tested using t-test and ANOVA, where applicable. The null hypothesis was rejected when the two-tailed $p < 0.05$ is indicated with *, <0.01 with **, and <0.001 with ***. The number of mice and hair cells is reported as *n*. At least five images were collected from 5 independent experiments for immunocytochemical experiments (Figs. 1, 2, 6). PCR experiments were repeated twice (Fig. 3), and all co-IP experiments were repeated at least in three independent experiments (Fig. 7).

### Reporting summary

Further information on research design is available in the Nature Portfolio Reporting Summary linked to this article.

## Data availability

The datasets generated during and analyzed during the current study are available as supplement data and the corresponding author upon request. Source data are provided in this paper. Source data are provided with this paper.

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

## Acknowledgements

We thank members of our laboratory for their comments on this manuscript and Dr. Ruth Anne Eatock and James Hudspeth for their constructive comments on the initial draft of the manuscript. This work was supported by grants to ENY from the National Institutes of Health (DC015135, DC016099, DC015252, AG051443, AG060504), S10 RR022498-01, P20GM130459, and supported, in part, by NIH R56 HL138392, R01 HL158961 and American Heart Association (AHA) 23SFRNPCS1060482, 23SFRNPCS1061606 (X.D.Z.), and NIH R01 HL085727, R01 HL085844, R01 HL137228, VA Merit Review Grant I01 BX000576 and I01 CX001490, AHA 23SFRNCCS1052478, 23SFRNPCS1061606, and 23SFRNPCS1060482 (N.C.), R01 HL159304, AG063796 (R.E.D.), NIH R56 HL167932, 23SFRNPCS1060482, and T32KT4729 from University of California Office Of President (P.S.), NIH F31 HL170698 (P.T.), and NIH F32 HL151130 (R.L.W.). Support for single-channel reconstitution and analyses came from NSF grant 1840842 to INP. The Harold S. Geneen Charitable Trust Awards Program for Coronary Heart Disease supported PNT.

## Author contributions

E.N.Y. designed the research; J.H.L., M.C.P.F., P.Th, P.Tr, X.-D.Z., I.N.P., B.S., and E.N.Y. analyzed data and wrote the manuscript; M.C.P.F., J.H.L., H.Y.K., Y.C., S.P., M.K., J.K., F.W., T.D., B.F., J.C., H.Z., G.P., P.Th, P.Tr, X.-.D.Z., R.E.D., N.C., and E.N.Y. performed the research. All authors read and approved the final manuscript.

## Competing interests

The authors declare no competing interests.
