## [Peer Review File · Nature Communications]

The Piezo channel is a mechano-sensitive complex component in the mammalian inner ear hair cellEditorial Note: This manuscript has been previously reviewed at another journal that is not operating a transparent peer review scheme. This document only contains reviewer comments and rebuttal letters for versions considered at *Nature Communications* .

REVIEWER COMMENTS

Reviewer #2 (Remarks to the Author):

In this paper from Yamoah and colleagues the authors make the remarkable claim that a heteromeric complex of Piezo1 and Piezo2 constitutes the pore forming core of the MET channel in mammalian hair cells. The claim is remarkable for several reasons. First, careful examination of the idea that Piezos constitute the MET channel have come to the opposite conclusion even showing that mice with Piezo1 and 2 conditional deletion in hair cells have only moderately elevated auditory brain stem responses (ABRs). Moreover, double deletion of Piezo1 and 2 did not lead to a greater effect than deletion of Piezo2 alone (see Wu et al PMID 27893727). The authors also make the claim that Piezo1 and 2 form a heteromeric ion channel in the hair cell. This latter claim is not supported by any direct evidence in this paper. Indeed, the authors do not actually show that deletion of Piezo1 or 2 in hair cells leads to deafness or loss of MET function instead their evidence is based on the miss-expression of mutant Piezo1 and 2 channels that are non-functional. Thus, expression of either non-functional variant in inner and outer hairs cells with two different promotors led to a substantial reduction in hearing accompanied later by hair cell degeneration. The authors include one control for overexpression of Piezo1 using the same genetic strategy and show that ABRs in these animals are similar to controls. Most other controls for all other experiments are animals carrying only the Cre-driver without driving transgene, which is not ideal. No control for overexpression of Piezo2 is shown which in principle is also necessary to interpret the results. This is because overexpression of the wild type protein (e.g. Piezo2) in the bundle using the same driver lines may lead to pathological changes by interfering with the localization or function of other proteins in the bundle. More characterization of the control Piezo1 or 2 overexpression mice would be necessary here, e.g. do they show hair cell degeneration later in life?

Although many of the results shown by the authors is very suggestive and interesting I do think that the authors conclusion that Piezo1 and 2 are central to the MET in the hair cell is not really supported by their data. Their evidence is based on the fact that Piezo1 and 2 are mechanosensitive channels present in the bundle and that pore dead variants suppress transduction. Yet, intrinsic mechanosensitivity of the MET channel is not even required for the MET channel as this complex is gated by a tether mechanism mediated by the tip links. In other words, the entire data set could easily be explained by some developmental or regulatory role of Piezo1 and 2 channels in the hair cell. The ideal experiment would be to introduce mutations that change Piezo1 or 2 biophysical properties and ask if this alters the native MET current in the predicted way. Strangely there are many known gain of function mutations in both Piezo1 and 2, but as far as I know none are associated with changes in hearing. Indeed, this human literature really should be discussed especially in light of the fact that virtually all components of the MET complex are associated with hearing disorders in both mice and humans.

The staining of tagged PIEZO1 and 2 proteins in the hair cells looks interesting, but how can the authors be sure that the tags can lead to mislocalization. Indeed the Piezo1tdT mouse has to my knowledge never been shown not to show abnormalities, but perhaps the authors can point to data on this.

All poking experiments were done in wild type Piezo1 N2A cells, if so why? N2A cells have endogenous Piezo1 as is well known from the Piezo cloning paper in 2010. The mutant proteins appear to block the endogenous current if that is the case. By transfecting cells with a mixture of Piezo1 and Piezo2 it is inevitable that currents with different kinetics will be seen, this cannot be interpreted as indicating the formation of heteromeric channels. This is just a mixed macroscopic current and the experiment does

not provide any evidence for a heteromeric Piezo1/2 channel.

A cryo structure of heteromeric Piezo channel has not been done this would be the only clear way to demonstrate the feasibility of such a configuration. I suggest that the authors omit any speculation about heteromeric Piezo1/2 channels as this is not even central to their story and there is really no evidence provided.

DHS only partially blocks the Piezos at huge concentrations as has already been shown (see PMID30473131), the doses that block the endogenous channel are much lower. The authors show that the IC50 for DHS block of the Piezo channel is dramatically lower when the channel is expressed with TMC1. This is a very intriguing result. Indeed, it seems to be the only robust effect of co-expression of Piezo with TMC1. The inactivation data with TMC1 and Piezos are hard to interpret in the light of the fact that N2A cells express endogenous Piezo1. It is also not clear if any of the data on inactivation of Piezos with TMC proteins is even statistically significant. Individual currents show a high degree of variability and it would be preference to use graphing methods to show this (individual data points).

The first section that attempts to claim that Piezo1 and 2 could have similar biophysical properties to the endogenous MET currents really feels like trying to get a round peg into a square hole. The inactivation properties of heterologously expressed Piezo cannot be compared to adaptation of MET currents *in vivo*. In addition, many properties of Piezo macroscopic currents are really not consistent with the MET channel. For example, current inactivation is highly asymmetric for Piezo channels so that currents hardly inactivate at positive potentials, but inactivate rapidly at negative potentials PMID: 29545531. This is not a feature of the MET current as far as I know.

In short this paper provides some novel data that will be of interest to the field but the authors are pushing a narrative that I do not think is supported by the data. They should tone down the interpretation of their data accordingly.

Reviewer #3 (Remarks to the Author):

The revised MS of Lee and colleagues presents a number of new experiments to address some of the concerns of the referees. It is good to see MET current recordings from OHCs included now, and also to see that the MET currents are further reduced in the Pz1 Pz2 double mutant KIs. These data, together with the nearly complete loss of ABR responses in the double mutant KIs by 8 weeks, provide strong evidence that in the presence of the mutated Piezo proteins nearly all the MET currents are abolished. However, crucially, this does not demonstrate that the Piezos themselves constitute the ion conducting pores of the MET channels. Very strong and growing evidence suggests that it is Tmc1 that fulfills that role. The results in the present MS are extremely interesting and intriguing and must be published. It is unfortunate, however, that the authors persist in overstating their findings to imply that the Piezos are the central pore protein of the MET channels, with the Tmcs as allosteric regulators.

The Piezo currents reported in the present MS in cell lines look like Piezo currents in various previous reports, and well as like the anomalous MET currents reported in auditory hair cells (proposed to be Piezo 2: reference 12 & 49 in the MS). They differ substantially in key aspects from the regular MET currents of auditory hair cells, and it is a pity that the authors did not engage with my comments on this discrepancy in my previous report. MET currents adapt, with the main time constants in the submillisecond to a few ms range in auditory hair cells (See e.g. Fettiplace & Kim, *Physiol Rev* 94: 951–986, 2014), whereas Piezo currents inactivate, with much slower time constants. Yet, the authors continue to refer to the Piezo currents as adapting throughout the MS. If the authors want to demonstrate that their Piezo currents adapt rather than inactivate, they should do experiments with an adapting step with superimposed test steps to distinguish between adaptation and inactivation (see e.g. Fig. 1 in Hacohen et al *J Neurosci* 1989 Nov;9(11):3988-97). Piezo currents are an order of magnitude less sensitive to block by the aminoglycoside antibiotic dihydrostreptomycin (DHS) that

auditory hair cells. The authors present new data which show that, when Tmc1 is co-expressed with Pz1 in the N2A cell line, the DHS block becomes two orders of magnitude stronger than that of auditory hair cells. This is a very interesting finding suggesting a strong interaction between Piezos and Tmcs, but this extremely strong block does not resemble the moderate block found in vivo for the auditory hair cells.

A number of assertions in the revised paper are too strong, vague, or go beyond traditional logic, presumably in trying to force the argument that the Piezos are the MET channel pores, for example:
End of abstract:

1. 'We affirm that Pz protein subunits constitute MET channels and that functional interactions with components of the MET complex yield current properties resembling hair-cell MET currents.' As stated above and in my previous report, the Piezo currents are not like regular MET currents.

2. 'Our results demonstrate Pz is a MET channel component central to interacting with MET complex proteins.' Too strong: 'is a MET channel component'. Vague: what does 'central to interacting with MET complex proteins' mean?

3. 'Results account for the MET channel pore and complex.' Vague.

Top of 3rd paragraph on page 4. The mere presence and co-localization of the Piezos along the stereocilia and cuticular plate does not demonstrate a (functional) role in the hair cells.

Top of page 5: 'Pz is central to the MET complex' – vague. The finding that Piezo interacts with various MET channel proteins does not imply 'a chief role in the MET complex assembly'. 'We propose that the MET complex comprises multimers of the Pz channel subunits serving as the centerpiece'. This is too strong: the Tmcs have a much stronger claim to that, with pore mutations and cysteine substitutions in the pore affecting the MET pore properties, including ion selectivity, DHS block and conductance.

Header to last paragraph of page 7: 'Properties of the Pz-channel current resemble the HC MET current.' They resemble the anomalous MET current reported by others (e.g. ref 11, 12, 49 in MS), but not the regular MET current of auditory hair cells.

Top of Discussion (page 18). I pointed out in my previous report: 'The use of a reference from 2009 to support the statement that 'the identity of the pore-forming protein remains unresolved' (start of Discussions page 13) is disingenuous.' Yet, this statement, and the reference, remain in the current version.

Bottom of page 18. 'Pz subunits form a non-specific cationic channel with biophysical and pharmacological features resembling the MET currents.' No, again, they resemble the anomalous MET current reported by others, but not the regular MET current.

End of first paragraph on page 19: 'Pz1/2 and Pz-Tmc co-expression yielded current with MET pharmacological characteristics, Ca²⁺-sensitivity, and permeation properties reminiscent of HC MET current'. Again, the resemblance to the regular MET current (as opposed to the anomalous MET current) is superficial and doesn't withstand scrutiny.

Page 19, 2nd paragraph. 'Moreover, IHCs and OHCs expressing Pz mutants have reduced FM1-43 uptake with attenuated HC transduction currents, substantiating the assertion that Pz is necessary for HC MET current.' Logically, if Pz were necessary for the MET current, there would be no FM1-43 uptake and no MET currents.

Other comments:

Page 7: The Pz1-Tmc2 and Pz2-Tmc1 NND distribution was $\sim 0.4 \pm 0.5 \mu\text{m}$ in IHC stereocilia and $\sim 0.5 \pm 0.4 \mu\text{m}$ in OHC stereocilia (Figs. 2f-h). Do the quoted numbers fit with the graphs? Most near neighbours seem to be at about 0.1 μm , not 0.4 or 0.5 μm .

Page 8: 'An inner-face pore radius of $7 \pm 4 \text{ \AA}$ and an outer-face radius of $20 \pm 5 \text{ \AA}$ was estimated (S7f-g). The findings align with HC MET channel⁵³ estimations and the upward concave structure of Pz⁴⁸.' Reference 58 meant (Farris et al 2004)?

Near end of page 9: 'Co-expression of Tmc1 or Tmc2 with mPz1 shifted displacement-response relationships rightward of IPz1 alone, with Tmc2 exerting the most pronounced effect (S9).' Was this shift significant? What was the criterion for declaring that the shift was larger for Tmc2 than for Tmc1? This is not evident from Fig S9f.

Page 13: 'Almost complete hearing loss (threshold $\geq 90 \text{ dB}$) was recorded for the double mutant mc-

Pz1-2MU mice by 8 weeks of age (Fig. 4b).¹ It would be useful to know whether this hearing loss was also progressive, like for the single KIs. What were the ABRs for the double KIs like at 4 weeks? Define RPI on first occurrence. 36 and 37 are the same reference.

REVIEWER COMMENTS

Reviewer #2 (Remarks to the Author):

In this paper from Yamoah and colleagues the authors make the remarkable claim that a heteromeric complex of Piezo1 and Piezo2 constitutes the pore-forming core of the MET channel in mammalian hair cells. The claim is remarkable for several reasons. First, careful examination of the idea that Piezos constitute the MET channel have come to the opposite conclusion even showing that mice with Piezo1 and 2 conditional deletion in hair cells have only moderately elevated auditory brain stem responses (ABRs). Moreover, double deletion of Piezo1 and 2 did not lead to a greater effect than deletion of Piezo2 alone (see Wu et al PMID 27893727).

Response:

We are grateful for the reviewer's cautionary suggestions that we should tone the report's conclusion down and be circumspect in the light of previous reports. We have made all suggested changes and included additional alterations to meet the recommendations.

Regarding Wu et al, PMID 27893727 (ref 48), we remain baffled by the report that the double Pz1/Pz2 mutation did not result in robust auditory and vestibular traits. We have offered an alternative explanation in the discussion for the modest phenotype (**Please see page 19, first paragraph**). We trust the discussion of the reviewer's concerns is adequate.

The authors also make the claim that Piezo1 and 2 form a heteromeric ion channel in the hair cell. This latter claim is not supported by any direct evidence in this paper. Indeed, the authors do not actually show that deletion of Piezo1 or 2 in hair cells leads to deafness or loss of MET function instead their evidence is based on the miss-expression of mutant Piezo1 and 2 channels that are non-functional. Thus, expression of either non-functional variant in inner and outer hairs cells with two different promoters led to a substantial reduction in hearing accompanied later by hair cell degeneration. The authors include one control for overexpression of Piezo1 using the same genetic strategy and show that ABRs in these animals are similar to controls. Most other controls for all other experiments are animals carrying only the Cre-driver without driving transgene, which is not ideal. No control for overexpression of Piezo2 is shown which in principle is also necessary to interpret the results. This is because overexpression of the wild type protein (e.g. Piezo2) in the bundle using the same driver lines may lead to pathological changes by interfering with the localization or function of other proteins in the bundle. More characterization of the control Piezo1 or 2 overexpression mice would be necessary here, e.g. do they show hair cell degeneration later in life?

This section of the reviewers is well noted. However, we provide a control set of data using Pz1 WT knockin. We agree that Pz2 WT knockin would also be gratifying. The report generates five new mouse models, Pz1 WT, knockin, Pz1 MU, Pz2 MU, Myosin 15 Cre, Calb2 Cre. Additionally, we included data from Pz1 and Pz2 knockout.

To date, we have not obtained any data from the Pz1 WT knockin model that shows hearing loss compared to WT mice, nor have we observed any hair cell degeneration that is different from that observed in C57 controls. We have stated it in Figure 4 legend. (**We have not observed any histological changes in the cc vs. Pz1^{WT} in 8-12-week-old mice, page 26**).

The data on Pz2 WT will be made available to the scientific community when ready and available.

Although many of the results shown by the authors is very suggestive and interesting I do think that the authors conclusion that Piezo1 and 2 are central to the MET in the hair cell is not really supported by their data. Their evidence is based on the fact that Piezo1 and 2 are mechanosensitive channels present in the bundle and that pore dead variants suppress transduction. Yet, intrinsic mechanosensitivity of the MET channel is not even required for the MET channel as this complex is gated by a tether mechanism mediated by the tip links. In other words, the entire data set could easily be explained by some developmental or regulatory role of Piezo1 and 2 channels in the hair cell. The ideal experiment would be to introduce mutations that change Piezo1 or 2 biophysical properties and ask if this alters the native MET current in the predicted way. Strangely there are many known gain of function mutations in both Piezo1 and 2, but as far as I know none are associated with changes in hearing. Indeed, this human literature really should be discussed especially in light of the fact that virtually all components of the MET complex are associated with hearing disorders in both mice and humans.

We have addressed the reviewer's concerns in the discussion as follows (see Page 22 last paragraph).

Limitations, and by no means extensive, of the current findings and the suggestion that the Pz channel constitutes a component of the MET complex are as follows. 1) Although a few gain-of-function of human *Pz1* and 2 mutations have been reported⁸⁶, none have reported altered auditory or vestibular hypofunction to date. As more studies in the fledgling field of Pz-channelopathy emerge, future findings may clarify such ambiguity. 2) Despite the potential for embryonic lethality, attempts to generate a biophysically predictable Pz mutant model would be gratifying to interrogate the MET current and auditory and vestibular functions. 3) Finally, although the current studies focused on HC MET channels, other potential Pz functions await future studies. For an emerging field of Pz channel functions, it is anticipated that impending discoveries will likely enhance and elucidate some of the uncertainties raised in the current findings.

The staining of tagged PIEZO1 and 2 proteins in the hair cells looks interesting, but how can the authors be sure that the tags can lead to mislocalization. Indeed the Piezo1tdT mouse has to my knowledge never been shown not to show abnormalities, but perhaps the authors can point to data on this.

We are unsure of the reviewer's concern. However, we did not observe any signs of labeled Pz mislocation. Additionally, we showed in **S16b** that in the Pz1MU mice, we could locate Pz at stereocilia tips.

All poking experiments were done in wild type Piezo1 N2A cells, if so why? N2A cells have endogenous Piezo1 as is well known from the Piezo cloning paper in 2010. The mutant proteins appear to block the endogenous current if that is the case. By transfecting cells with a mixture of Piezo1 and Piezo2 it is inevitable that currents with different kinetics will be seen, this cannot be interpreted as indicating the formation of heteromeric channels. This is just a mixed macroscopic current and the experiment does not provide any evidence for a heteromeric Piezo1/2 channel.

A cryo structure of heteromeric Piezo channel has not been done this would be the only clear way to demonstrate the feasibility of such a configuration. I suggest that the authors omit any speculation about heteromeric Piezo1/2 channels as this is not even central to their story and there is really no evidence provided.

The reported data on recordings from N2A cells and the endogenous MA current (Page 8) agree well with the reviewer's report. Our previous response explained why we used N2A cells in the "poking" experiments. Not only were N2A cells robust for the "poking" experiments, but also the expression of Tmc in N2A cells (S6) suggested ~ 10% protein membrane trafficking. These were the motivations for using N2A cells.

We agree that co-expression of Pz1 and Pz2 and the resulting altered current does not indicate heterotrimeric interaction. In the previous reviews, we were chastised on this issue, and at the time, it was argued that because the sequence homology between Pz1 and Pz2 was ~42%, they were unlikely to interact.

In the revised MS, we examined the Pz1 and Pz2 sequences (**S2-S3**) and showed that at the Pz1 (crystal structure) subunit interface, Pz1 and Pz2 share ~84% sequence homology similar to K (SK) channels, in which structural and functional interaction has been demonstrated. While the ultimate experiments from cryoEM/crystal structural data await the future, we strongly disagree that the current data prevents us from any reasonable speculation. We only suggested the two channels may form heterotrimers.

DHS only partially blocks the Piezos at huge concentrations as has already been shown (see PMID30473131), the doses that block the endogenous channel are much lower. The authors show that the IC50 for DHS block of the Piezo channel is dramatically lower when the channel is expressed with TMC1. This is a very intriguing result. Indeed, it seems to be the only robust effect of co-expression of Piezo with TMC1. The inactivation data with TMC1 and Piezos are hard to interpret in the light of the fact that N2A cells express endogenous Piezo1. It is also not clear if any of the data on inactivation of Piezos with TMC proteins is even statistically significant. Individual currents show a high degree of variability and it would be preference to use graphing methods to show this (individual data points).

We thank the reviewer for the comments on the data regarding DHS differential effects on Pz and Pz/TMC currents. In response to the reviewer's comments, we have added additional data and plotted the inactivation of individual data points as requested in Figure **S9**. In the figure legend, we provide details of the statistical comparisons, which were presented in the previous revised version but were missed somehow.

The first section that attempts to claim that Piezo1 and 2 could have similar biophysical properties to the endogenous MET currents really feels like trying to get a round peg into a square hole. The inactivation properties of heterologously expressed Piezo cannot be compared to adaptation of MET currents in vivo. In addition, many properties of Piezo macroscopic currents are really not consistent with the MET channel. For example, current inactivation is highly asymmetric for Piezo channels so that currents hardly inactivate at positive potentials, but inactivate rapidly at negative potentials PMID: 29545531. This is not a feature of the MET current as far as I know.

We recognize the reviewer's comments on adaptation and inactivation. We agree that the decay of the Pz current is more consistent with inactivation than adaptation. We have, therefore, called the Pz current either inactivation or decay. We apologize for the error.

In short this paper provides some novel data that will be of interest to the field but the authors are pushing a narrative that I do not think is supported by the data. They should tone down the interpretation of their data accordingly.

We thank the reviewer for the cautionary counsel. We have modified the MS accordingly.

Reviewer #3 (Remarks to the Author):

The revised MS of Lee and colleagues presents a number of new experiments to address some of the concerns of the referees. It is good to see MET current recordings from OHCs included now, and also to see that the MET currents are further reduced in the Pz1 Pz2 double mutant KIs. These data, together with the nearly complete loss of ABR responses in the double mutant KIs by 8 weeks, provide strong evidence that in the presence of the mutated Piezo proteins nearly all the MET currents are abolished. However, crucially, this does not demonstrate that the Piezos themselves constitute the ion conducting pores of the MET channels. Very strong and growing evidence suggests that it is Tmc1 that fulfils that role. The results in the present MS are extremely interesting and intriguing and must be published. It is unfortunate, however, that the authors persist in overstating their findings to imply that the Piezos are the central pore protein of the MET channels, with the Tmcs as allosteric regulators.

The Piezo currents reported in the present MS in cell lines look like Piezo currents in various previous reports, and well as like the anomalous MET currents reported in auditory hair cells (proposed to be Piezo 2: reference 12 & 49 in the MS). They differ substantially in key aspects from the regular MET currents of auditory hair cells, and it is a pity that the authors did not engage with my comments on this discrepancy in my previous report. MET currents adapt, with the main time constants in the submillisecond to a few ms range in auditory hair cells (See e.g. Fettiplace & Kim, *Physiol Rev* 94: 951–986, 2014), whereas Piezo currents inactivate, with much slower time constants. Yet, the authors continue to refer to the Piezo currents as adapting throughout the MS. If the authors want to demonstrate that their Piezo currents adapt rather than inactivate, they should do experiments with an adapting step with superimposed test steps to distinguish between adaptation and inactivation (see e.g. Fig. 1 in Hacoen et al *J Neurosci* 1989 Nov;9(11):3988-97). Piezo currents are an order of magnitude less sensitive to block by the aminoglycoside antibiotic dihydrostreptomycin (DHS) than auditory hair cells. The authors present new data which show that, when Tmc1 is co-expressed with Pz1 in the N2A cell line, the DHS block becomes two orders of magnitude stronger than that of auditory hair cells. This is a very interesting finding suggesting a strong interaction between Piezos and Tmcs, but this extremely strong block does not resemble the moderate block found in vivo for the auditory hair cells.

Similar issues were raised by reviewer 2 regarding current inactivation, adaptation, and Pz vs. Pz/Tmc current sensitivity to the DHS. We have made the suggested changes and have toned

the conclusions down. Whether Pz- and/or Pz/Tmc-current adapt is beyond this report's scope and will be assigned to future studies.

A number of assertions in the revised paper are too strong, vague, or go beyond traditional logic, presumably in trying to force the argument that the Piezos are the MET channel pores, for example:

End of abstract:

1. 'We affirm that Pz protein subunits constitute MET channels and that functional interactions with components of the MET complex yield current properties resembling hair-cell MET currents.' As stated above and in my previous report, the Piezo currents are not like regular MET currents.

We have replaced the statement above: We suggest that Pz protein subunits constitute part of the MET complex and that interactions with other MET complex components yield functional MET units to generate hair-cell MET currents (see abstract).

2. 'Our results demonstrate Pz is a MET channel component central to interacting with MET complex proteins.' Too strong: 'is a MET channel component'. Vague: what does 'central to interacting with MET complex proteins' mean?

We have removed the strong assertion as recommended.

3. 'Results account for the MET channel pore and complex.' Vague.

Top of 3rd paragraph on page 4. The mere presence and co-localization of the Piezos along the stereocilia and cuticular plate does not demonstrate a (functional) role in the hair cells.

We have removed the statement.

Top of page 5: 'Pz is central to the MET complex' – vague. The finding that Piezo interacts with various MET channel proteins does not imply 'a chief role in the MET complex assembly'. 'We propose that the MET complex comprises multimers of the Pz channel subunits serving as the centerpiece'. This is too strong: the Tmcs have a much stronger claim to that, with pore mutations and cysteine substitutions in the pore affecting the MET pore properties, including ion selectivity, DHS block and conductance.

We have made the changes as recommended e.g. " Pz may be part of the MET complex and interacts with multiple proteins, including Pcdh15, Tmc1, Lhfpl5, Tmie, and Cib2, suggesting a role in the MET complex assembly. We propose the MET complex includes the Pz channel subunits that serve along with other regulatory and indispensable binding partners such as TMC to confer a functional MET unit.

".

Header to last paragraph of page 7: 'Properties of the Pz-channel current resemble the HC MET current.' They resemble the anomalous MET current reported by others (e.g. ref 11, 12, 49 in MS), but not the regular MET current of auditory hair cells.

The suggestion is well taken, and the header title is altered accordingly. "**Properties of the Pz-channel current resemble the HC anomalous MET current**".

Top of Discussion (page 18). I pointed out in my previous report: 'The use of a reference from 2009 to support the statement that 'the identity of the pore-forming protein remains unresolved' (start of Discussions page 13) is disingenuous.' Yet, this statement, and the reference, remain in the current version.

We have adjusted that statement accordingly and provided references consistent with the first and second paragraphs in the discussions.

Bottom of page 18. 'Pz subunits form a non-specific cationic channel with biophysical and pharmacological features resembling the MET currents.' No, again, they resemble the anomalous MET current reported by others, but not the regular MET current.

We have removed all suggestions that the Pz current resembles the MET current. Instead, we have drawn parallels between the Pz and anomalous currents.

End of first paragraph on page 19: 'Pz1/2 and Pz-Tmc co-expression yielded current with MET pharmacological characteristics, Ca²⁺-sensitivity, and permeation properties reminiscent of HC MET current'. Again, the resemblance to the regular MET current (as opposed to the anomalous MET current) is superficial and doesn't withstand scrutiny.

We have removed all suggestions that the Pz current resembles the MET current. Instead, we have drawn parallels between the Pz and anomalous currents.

Page 19, 2nd paragraph. 'Moreover, IHCs and OHCs expressing Pz mutants have reduced FM1-43 uptake with attenuated HC transduction currents, substantiating the assertion that Pz is necessary for HC MET current.' Logically, if Pz were necessary for the MET current, there would be no FM1-43 uptake and no MET currents.

We respectfully disagree. Introducing a non-functional channel at the ROSA 26 locus does not entirely abolish the endogenous channel. Thus, a residual current and low FM1-43 uptake should not be a surprise. We have made this statement in the revised version, and **S23** points to our reasoning.

Other comments:

Page 7: The Pz1-Tmc2 and Pz2-Tmc1 NND distribution was $\sim 0.4 \pm 0.5 \mu\text{m}$ in IHC stereocilia and $\sim 0.5 \pm 0.4 \mu\text{m}$ in OHC stereocilia (Figs. 2f-h). Do the quoted numbers fit with the graphs? Most near neighbours seem to be at about 0.1 μm , not 0.4 or 0.5 μm .

We have re-analyzed the data and provided the NND reflecting the data shown. We apologize for the oversight.

Page 8: 'An inner-face pore radius of $7 \pm 4 \text{ \AA}$ and an outer-face radius of $20 \pm 5 \text{ \AA}$ was estimated (S7f-g). The findings align with HC MET channel⁵³ estimations and the upward concave structure of Pz⁴⁸.' Reference 58 meant (Farris et al 2004)?

We have cited the correct reference (Farris et al., 2004).

Near end of page 9: 'Co-expression of Tmc1 or Tmc2 with mPz1 shifted displacement-response relationships rightward of IPz1 alone, with Tmc2 exerting the most pronounced effect (S9).' Was this shift significant? What was the criterion for declaring that the shift was larger for Tmc2 than for Tmc1? This is not evident from Fig S9f.

This section of the report received mixed reviews from two reviewers. Because of the coarse nature of the stimulus probe, one reviewer suggested it is not appropriate to compare the displacement response shifts quantitatively. It is for this reason that we refrained from stating the statistical comparison. Instead, we have stated the comparison was qualitative.

Page 13: 'Almost complete hearing loss (threshold ≥ 90 dB) was recorded for the double mutant mc-Pz1-2MU mice by 8 weeks of age (Fig. 4b).' It would be useful to know whether this hearing loss was also progressive, like for the single KIs. What were the ABRs for the double KIs like at 4 weeks?

The requested information was provided in the legend of S17. The 8-week-old thresholds were higher than the 4-week-old Pz1-2MU mice.

Define RPI on first occurrence.
36 and 37 are the same reference.

RPI is defined on the first occurrence, and the reference duplication has been fixed (Page 4).

REVIEWERS' COMMENTS

Reviewer #3 (Remarks to the Author):

This third version of the MS by Lee and colleagues is much improved and presents valuable and noteworthy findings on contributions of Piezo channels to hair-cell mechanotransduction. In particular the title, summary and introduction represent their findings in a more balanced and considered manner than before. The concerns in my previous reviews about referring to the Piezo currents in the N2A cells as adapting rather than inactivating have also been addressed. The discussion is improved, but some issues remain. It is hard to explain the details of the findings in the Pan et al study by assuming TMC is an allosteric regulator (line 530-532). You would have to explain altered calcium selectivity and DHS block of the permeation pore by allosteric modification. You also have to explain that the MTS reagents, when they are applied when the MET channel is closed, do not bind to the cysteine-modified residues in TMC1, and do not block the MET currents. These are strong arguments for TMC forming the MET channel pore, which the findings in the current MS do not falsify. The evidence for the Piezos being the MET channel pore is circumstantial and less direct than the evidence for TMC1, but your study points the way for future experiments to address this. An acknowledgement of this fact should be added to the list of limitations that you provide at the end of the discussion (line 542-551). I also suggest you remove the third point (line 548-551) which is vague and questionable: there are lots of studies on Piezo function in mechanosensation in systems not related to hearing and balance.

Minor issues:

1. Line 252 – 'located spatially' is tautological. Use 'colocalized' instead, if that is what is meant?
2. Line 254. The header of the section on pharmacology is too broad and suggests that much more work was done to test similarities between the piezo and MET currents. For example, you did not check for permeant block, or test other MET channel blockers. The only similarity you found is that, like MET currents, IPz1 and IPz1/T1 are blocked by DHS. The IC50s of each of these don't match that of the MET current, one being >20-fold less sensitive and the other >100-fold more sensitive. To capture the essence of your findings I suggest as header: 'Co-transfection of TMC1 increases DHS block of IPz1 by 3 orders of magnitude'.
3. Line 261. Remove 'and its' between 'Tmc' and 'mutations'.
4. Line 263. Sensitivity to DHS is increased nearly 4000 fold by coexpressing TMC1, rather than 1000-fold as you state.
5. Line 393. You would expect the controls to be intact. Better header: 'OHC electromotility in PzMU mutant mice was relatively intact compared to controls'
6. Line 444-446. 'Experiments implicating.....another channel29'. Unclear, incomplete sentences – what is meant here?
7. Line 521-522. Modify this sentence, which seems to imply that TMC is not thought to form an ion channel in vertebrates, even though reference 29 states that it forms the MET channel pore.
8. Line 528. Replace 'mediate' by 'alter'. Not clear what mediate means in this context.

Corné Kros.

Reviewer #3 (Remarks to the Author):

This third version of the MS by Lee and colleagues is much improved and presents valuable and noteworthy findings on contributions of Piezo channels to hair-cell mechanotransduction. In particular, the title, summary and introduction represent their findings in a more balanced and considered manner than before. The concerns in my previous reviews about referring to the Piezo currents in the N2A cells as adapting rather than inactivating have also been addressed. The discussion is improved, but some issues remain. It is hard to explain the details of the findings in the Pan et al study by assuming TMC is an allosteric regulator (line 530-532). You would have to explain altered calcium selectivity and DHS block of the permeation pore by allosteric modification. You also have to explain that the MTS reagents, when they are applied when the MET channel is closed, do not bind to the cysteine-modified residues in TMC1, and do not block the MET currents. These are strong arguments for TMC forming the MET channel pore, which the findings in the current MS do not falsify. The evidence for the Piezos being the MET channel pore is circumstantial and less direct than the evidence for TMC1, but your study points the way for future experiments to address this. An acknowledgement of this fact should be added to the list of limitations that you provide at the end of the discussion (line 542-551). I also suggest you remove the third point (line 548-551) which is vague and questionable: there are lots of studies on Piezo function in mechanosensation in systems not related to hearing and balance.

Response:

We have addressed the report's limitations as raised by the reviewer in the discussion section (**lines 540-553.....** 1) Suppose Tmc is an allosteric regulator of Pz; what is difficult to explain is the altered Ca²⁺ selectivity and DHS channel-pore block by allosteric modification and the implicit suggestion of the methanethiosulfonate (MTS) reagents effect on Tmc function is state-dependent²⁹ (page 22)..... Additionally, we have removed the previous third point, as recommended by the reviewer.

Minor issues:

1. Line 252 – 'located spatially' is tautological. Use 'colocalized' instead, if that is what is meant?

Response:

We have replaced "located spatially" with "colocalized" (**page 10 line 252**).

2. Line 254. The header of the section on pharmacology is too broad and suggests that much more work was done to test similarities between the piezo and MET currents. For example, you did not check for permeant block, or test other MET channel blockers. The only similarity you found is that, like MET currents, IPz1 and IPz1/T1 are blocked by DHS. The IC50s of each of these don't match that of the MET current, one being >20-fold less sensitive and the other >100-fold more sensitive. To capture the essence of your findings I suggest as header: 'Co-transfection of TMC1 increases DHS block of IPz1 by 3 orders of magnitude'.

Response:

We have changed the header as suggested and have indicated the differences in sensitivity of the MET current and IPz1 and IPz1/T1 (**page 10 line 254**).

3. Line 261. Remove 'and its' between 'Tmc' and 'mutations'.

Response:

We have removed "and its" (**page 10, line 261**).

4. Line 263. Sensitivity to DHS is increased nearly 4000 fold by coexpressing TMC1, rather than 1000-fold as you state.

Response

We have fixed the typographical error from 1000 to 4000-fold (**page, 11 line 263**).

5. Line 393. You would expect the controls to be intact. Better header: 'OHC electromotility in PzMU mutant mice was relatively intact compared to controls.'

Response

We have changed the header to 'OHC electromotility in Pz^{MU} mutant mice was relatively intact compared to controls (**page, 16 line 392**).'

6. Line 444-446. 'Experiments implicating.....another channel²⁹'. Unclear, incomplete sentences – what is meant here?

Response

We have re-phrased the sentence: "Experiments implicating Tmc1 as a pore-forming protein in HCs did not consider Tmc's allosteric role on another channel²⁹" (**page 18, line 443-444**).

7. Line 521-522. Modify this sentence, which seems to imply that TMC is not thought to form an ion channel in vertebrates, even though reference 29 states that it forms the MET channel pore.

Response

We have re-stated the sentence: "Tmc is an integral element of the MET complex in mammalian HCs and may serve as an ion channel^{29,36}". (**page 21 lines 519-522**)"

8. Line 528. Replace 'mediate' by 'alter'. Not clear what mediate means in this context.

Response

We have replaced "mediate" with "alter" (**page 21 line 526**).